# Higher interglacial dust fluxes relative to glacial periods in southwestern North American deserts

Spencer E. Staley [1,2] ✉, Peter J. Fawcett [1], R. Scott Anderson [3] & Matthew E. Kirby [4]

Lengthy and continuous dust flux records from the deserts of southwestern North America are needed to understand the long-term impacts of climate on dust emissions and the relative influence of regional versus intercontinental dust sources. Here, we perform a particle size end-member analysis of a ~230,000-year sediment record from Stoneman Lake, Arizona, USA, revealing the area directly downwind of this important dust source region is 1.2–10 times dustier during peak interglacial periods compared to glacial maxima. This contrasts with other global dust records and aligns with the region's distinctive dry interglacial and pluvial glacial climates. Our analyses indicate minimal deposition of intercontinental dust and suggest the primary driver of dustiness is regional fine sediment supply, governed by desert alluvial and fluvial system responses to climate change. On the other hand, changes in atmospheric circulation and factors affecting wet and dry fallout appear to exert little control on regional dust flux variability.

The Basin and Range province of southwestern North America (SWNA) is a large desert region and a major source of atmospheric mineral dust[1]. Situated between the Cascade-Sierra Nevada-Transverse mountain system and the Colorado Plateau (Fig. 1a, b), this region has emitted considerable amounts of dust since at least the mid-Pleistocene[2]. Yet, continuous records of dust deposition downwind scarcely extend beyond the Holocene (e.g., refs. 3,4). This lack of long-term data hinders efforts to contextualize recent dust increases, many of which are attributed to anthropogenic disturbance[5–7]. Because atmospheric dust reflects upwind geomorphic and land-surface processes, dust deposition records offer a powerful proxy for reconstructing how landscapes respond to climate over long timescales—insights that are often inaccessible through instrumental or historical observation alone[8].

This data gap also limits insight into how dust fluxes in SWNA respond to glacial-interglacial climate change—an important concern given that pluvial conditions and limited glaciogenic dust sources during glacial periods potentially drive regional eolian dynamics to differ from broader global trends. Because dust is both a product and a driver of Earth system processes, long-term records inform a wide range of environmental inquiry. Dust plays a crucial role in Earth's climate, impacting radiative balance, cloud formation, precipitation, and nutrient cycles[9–11]. Reconstructing dust fluxes from SWNA can help quantify the region's role in broader Earth system feedbacks, including its influence on North American ecosystems relative to dust from intercontinental sources[12] and its contribution to dust deposition in Greenland and surrounding oceans[13]. Such records also support improved calibration of soil chronosequences[14], refinement of soil production and transport models[15], and validation of climate models that simulate dust–climate feedbacks[16,17].

This study presents evidence that dust emissions from SWNA are out of phase with global patterns over the last two glacial-interglacial cycles. We reconstruct regional dust flux by calculating the mass accumulation rates of particle size end members in a 230 kyr-long lacustrine sediment core record from Stoneman Lake, Arizona (2050 m elevation, 34.779 N, 111.518 W). Situated on the southwestern

[1]Department of Earth & Planetary Sciences, University of New Mexico, Albuquerque, NM, USA. [2]Department of Earth and Ecosystem Sciences, Desert Research Institute, Reno, NV, USA. [3]School of Earth and Sustainability, Northern Arizona University, Flagstaff, AZ, USA. [4]Department of Geological Sciences, California State University, Fullerton, CA, USA. ✉e-mail: spencer.staley@dri.edu

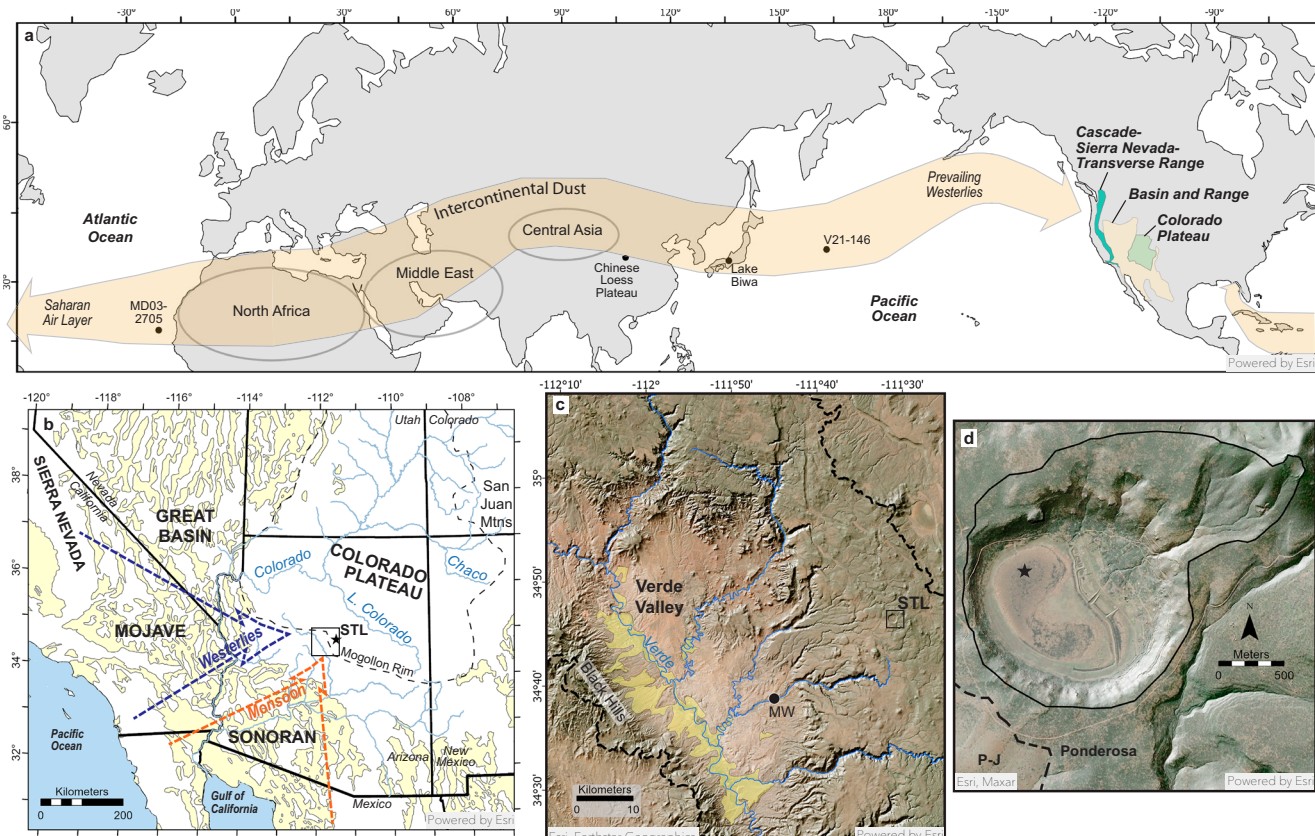

**Fig. 1 | Stoneman Lake, Arizona, USA: location, dust sources, and pathways. a** Major Northern Hemisphere dust sources (gray circles) and their atmospheric pathways to North America[12,19,21,71] (shaded orange arrows). Intercontinental dust travels via westerly air currents from Africa, the Middle East, and Central Asia across the Pacific Ocean. African dust traveling eastward via the Saharan Air Current primarily affects eastern North America. Dots indicate locations of dust accumulation records referenced in Fig. 4a[72–75]. **b** A regional map of southwestern North America with likely dust sources of unconsolidated Quaternary deposits[76], including alluvium, playas, and floodplains (shaded yellow). Dust emitted from regional sources is transported onto the Colorado Plateau by prevailing westerly and southerly monsoonal air circulations[32–34] (dashed arrows). Stoneman Lake (star, labeled STL) is located on the southwestern edge of the Colorado Plateau (dashed black). Box outlines area of (**c**), the Verde Valley. Tributaries of the southeast-flowing Verde River (watershed boundary indicated by dashed line) drain this portion of the Colorado Plateau, the Mogollon Rim. The broad valley contains extensive Quaternary fluvial and piedmont deposits[77] (shaded yellow) containing fine-grained sediments. It also features another lacustrine dust record, from Montezuma Well (MW)[4]. The black box outlines the area of (**d**), a hillshade relief map overlain on aerial imagery of the Stoneman Lake catchment (black outline) and the nearby ponderosa–piñon-juniper ecotone (dashed line). Core STL14 was recovered from the lake's northern depocenter (star). Map services and data available from the U.S. Geological Survey, National Geospatial Program.

margin of the Colorado Plateau, this enclosed, small (3.5-km²; ~5.5 catchment-to-lake area ratio) and steep (~12.6° average) alluvial basin with a basaltic catchment serves as an ideal natural dust trap, strategically located directly downwind of the Basin and Range province (Fig. 1). We compare these data to records of regional climate and landscape change to evaluate key drivers of dust flux across SWNA.

## Results

### Laser granulometry and mineralogy

Laser granulometric analyses of the upper ~13 m of core STL14 ($n = 205$) indicate that sediments in Stoneman Lake predominantly consist of silt, with smaller fractions of clay and sand, similar to the particle size distributions of soils in the local catchment ($n = 6$) (Fig. 2a). While most lake sediment samples exhibit modes within the clay and fine silt ranges, occasional modes appear in the coarse silt, fine sand, and coarse sand ranges. Detailed descriptions of sampling procedures and particle size analyses are provided in the "Methods" section.

Previous mineralogic studies have established that atmospheric dust comprises a large portion of sediments in Stoneman Lake[18]. X-ray diffraction analysis reveals an average mineral composition of 51% illite, 28% quartz, 9% kaolinite, 8% albite, 2% montmorillonite, 2% zircon, and 1% ilmenite. In contrast, local alkali basalt bedrock is composed of 40% pyroxene, 39% albite+anorthite, 17% olivine, 3% nepheline, and 1% ilmenite. The presence of illite, quartz, and zircon—not derived from local bedrock—indicates substantial inputs from external dust sources.

### End-member modeling and source attribution

A non-parametric end-member (EM) analysis using five end members optimally captures the variance within the lake sediment particle size distribution dataset ($r^2 = 0.9891$) while minimizing intercorrelations between end-member abundances (Supplementary Figs. 2a, b and 4). Detailed descriptions of modeling approaches and comparisons are provided in the "Methods" section. EM1 is polymodal, consisting mostly of clay to silt-sized grains with modes at 0.18, 3.0, 27, and 600 µm (Fig. 2b). EM2 is unimodal, composed of clay to fine silt-sized grains with a mode of 5.3 µm. EM3, also unimodal, consists mostly of silt-sized grains with a mode of 13 µm. EM4 comprises silt to very fine sand with a mode of 38 µm. EM5 is polymodal, containing mostly sand-sized grains with modes at 0.18, 17, and 169 µm. Mass accumulation rates for each end member are calculated based on their fractional abundance (Fig. 3a), density, and sedimentation rate (see "Methods" section).

We attribute these end members to either local alluvial or external dust sources using sedimentological analysis, transport dynamics, comparisons with previous studies, and correlations with geochemical

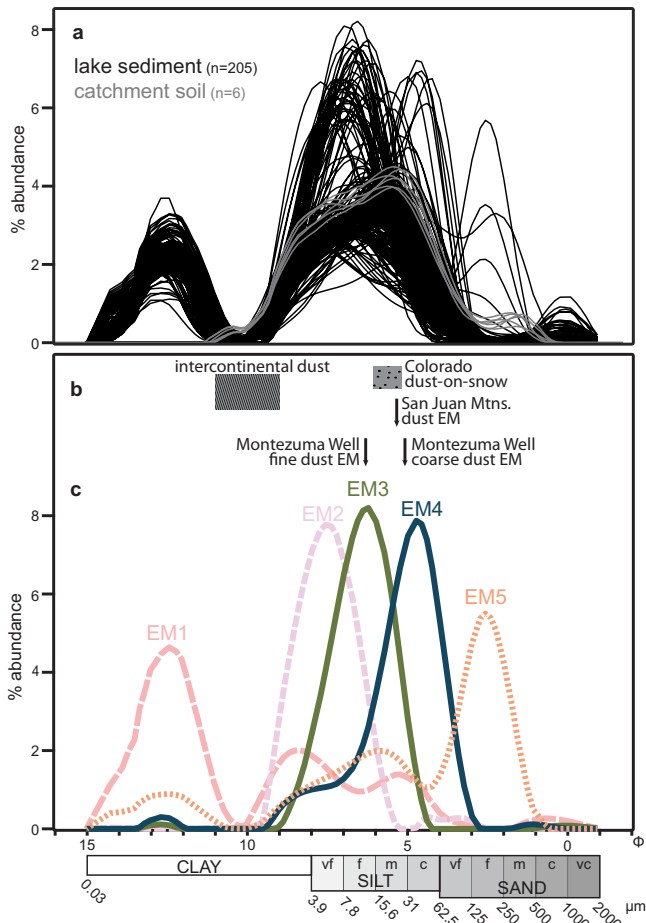

Fig. 2 | **Particle size and end-member analyses. a** Grain size distributions of Stoneman Lake catchment soil (gray) and lake sediment (black). **b** Grain size ranges (boxes) and end-member (EM) modes (arrows) of intercontinental[19–21] and regional dusts[3,4,22]. **c** Particle size distributions of clastic sediment end members generated by a non-parametric end-member model of lake sediment particle size data in (**a**). EM1 (modes: 0.18, 3.0, 27, and 600 µm; 33.4% of total clastic sediment volume), EM2 (mode: 5.3 µm; 17.9%), and EM5 (modes: 0.18, 17, and 169 µm; 2.4%) represent clastic sediment derived from basalt of the local lake catchment (shades of pink). EM3 (mode: 13 µm; 27.6%) corresponds to fine-grained dust (green). EM4 (mode: 38 µm; 18.8%) corresponds to coarse-grained dust (dark blue).

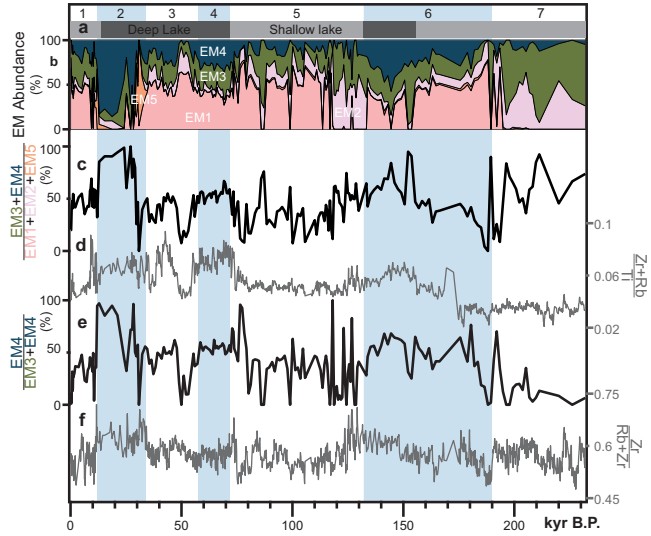

Fig. 3 | **End-member abundances and geochemical indicators in lake sediments.** Glacial-interglacial cycles numbered by Marine Isotope Stage (strong glacials shaded blue). **a** Relative Stoneman Lake depth[45]. **b** Percent abundance of five clastic sediment end members (EM) present in the Stoneman Lake core. **c** Ratio of dust end members (EM3 and EM4) to locally derived clastic end members (EM1, EM2, and EM5). **d** Geochemical ratio indicating the amount of dust, represented by Rb (from illite) and Zr (from zircon), relative to locally derived clastics, represented by Ti (from ilmenite), in core sediments[18]. Note correspondence to (**c**). **e** Ratio of coarse dust end member (EM4) to total dust (EM3 and EM4). **f** Geochemical ratio indicating the amount of coarse dust, represented by Zr, relative to total dust, represented by Rb and Zr[18]. Note correspondence to (**e**).

tracers. EM1, EM2, and EM5 collectively represent locally derived clastic sediments, accounting for 54% of the total clastic sediment volume. EM1 and EM5 exhibit polymodal distributions indicative of poor sorting, consistent with local origins. Although EM2 could be interpreted as intercontinental dust, its modal grain size (5.3 µm), accumulation rate, and variability are inconsistent with it coming from a transoceanic source. Observational data indicate that most intercontinental dust reaching western North America is likely to be in the range of 0.5–3 µm[19–21]. Furthermore, EM2 accumulation rates (Supplementary Fig. 3c) both exceed and are out of phase with measurements downwind of potential intercontinental dust sources, which peak during glacial periods (Fig. 4a). Instead, EM2 accumulates most rapidly as local lake levels fall during early interglacial periods making it more consistent with lake margin sediment redeposited into the center of the basin (e.g., ref. 3).

EM3 (mode: 13 µm) and EM4 (mode: 38 µm) are interpreted as dust, comprising 46% of the total clastic sediment volume (Fig. 2c). Their particle size distributions are too large to be interpreted as intercontinental dust and instead closely resemble other modern and paleo-dust deposits on the Colorado Plateau[3,4,22] (Fig. 2b). A nearby

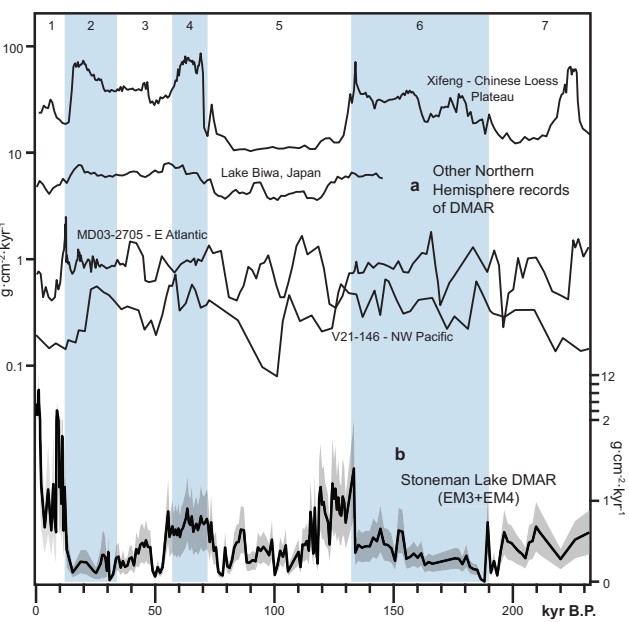

Fig. 4 | **Glacial-interglacial dust records: Northern Hemisphere.** Glacial-interglacial cycles numbered by Marine Isotope Stage (strong glacials shaded blue). **a** Other long dust mass accumulation rate (DMAR) records from the Northern Hemisphere. Dust records (locations indicated in Fig. 1a) include the Chinese Loess Plateau at Xifeng, China[72]; Lake Biwa, Japan[73]; E. Atlantic core MD03-2705 downwind of Saharan West Africa[74]; and Northwest Pacific core V21-146[75]. **b** Total dust mass accumulation rate at Stoneman Lake. Error envelope based on percent error propagation of values in the mass accumulation rate calculation (see "Methods" section).

study of Holocene sediments from Montezuma Well, Arizona, identified two dust end members with nearly identical modes to EM3 and EM4[4] (Fig. 2b). Montezuma Well is in a small sinkhole in carbonate bedrock located only 25 km WSW of Stoneman Lake and is likely downwind of the same dust sources (Fig. 1c).

Mineralogical and geochemical analyses offer further support for source attributions. Previous measurements of conservative elements Rb, Zr, and Ti serve as tracers for eolian and local sedimentary inputs[18]. Rb, substituting for K as an interlayer cation in illite clay ((K,H$_3$O)(Al,Mg,Fe)$_2$(Si,Al)$_4$O$_{10}$[(OH)$_2$,(H$_2$O)]) and Zr, controlled by zircon (ZrSiO$_4$) abundance, together represent dust input. Because zircon grains are larger and heavier than illite clay particles, the Zr to Rb ratio is an indicator of the relative grain size of dust inputs. Local inputs are

indicated by Ti, here mainly associated with ilmenite (FeTiO$_3$)[18]. (Zr+Rb)/Ti correlates positively with the abundance of EM3 and EM4 relative to EM1, EM2, and EM5 (Fig. 3c, d), reinforcing the interpretation that EM3 and EM4 are dust and EM1, EM2, and EM5 collectively represent local alluvial sediment. Additionally, the EM4/(EM3 + EM4) ratio positively correlates with Zr/(Rb+Zr) (Fig. 3e, f), supporting the distinction of EM4 as coarse dust and EM3 as fine dust.

## Sediment fluxes

Local sediment accumulation rate (EM1 + EM2 + EM5), or erosional flux, rises sharply after glacial terminations and remains elevated until the next glacial maxima (Fig. 5d). Median erosional flux is 4.72 g·cm$^{-2}$·kyr$^{-1}$ during interglacials and 2.45 g·cm$^{-2}$·kyr$^{-1}$ during glacials

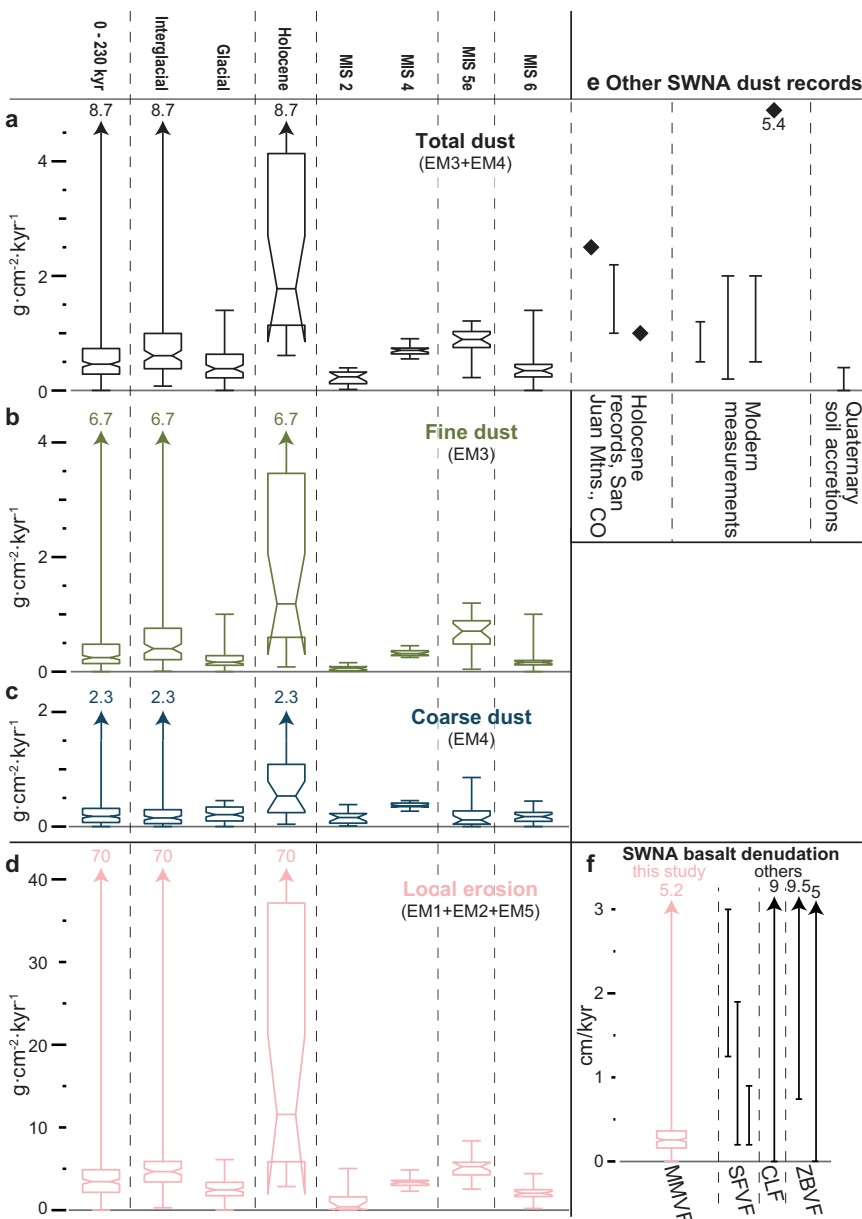

**Fig. 5 | Dust and erosion rates at Stoneman Lake and across Southwestern North America (SWNA).** Mass accumulation rate box plots for **a** total dust, **b** fine dust, **c** coarse dust, and **d** locally eroded sediments during intervals of the last 230 kyr at Stoneman Lake, Arizona. Interglacial includes all data from the Holocene, Marine Isotope Stage (MIS) 5, and MIS 7. Glacial includes all data from MIS 2, 4, and 6. **e** Dust fluxes measured at other sites in southwestern North America[3,14,23–28]. Note, (**a**), (**b**), (**c**), and (**e**) share the same y-axis scale. **f** denudation rates of basalts on the Colorado Plateau in Arizona and New Mexico. Box plot of denudation rates at Stoneman Lake (red) over the period of record derived from the local erosion rate and denudation rate ranges measured on other basalts (black), including the San Francisco Volcanic Field[78–80] (SFVF), Carrizozo Lava Flows[81] (CLF), and the Zuni-Banderas Volcanic Field[82,83] (ZBVF). In all box plots, the central line is the median, and where notches of two box plots do not overlap, their medians differ with 95% confidence. The bottom and top edges of the box indicate the 25th and 75th percentiles, and whiskers define the range including outliers.

(Fig. 5d and 6h). The median denudation rate over the last 230 kyr is 0.35 cm/kyr (derived directly from erosional flux, see "Methods" section), consistent with other regional basalt flows (Fig. 5f).

Median total dust accumulation rate (EM3 + EM4), or dust flux, over the last 230 kyr is 0.46 g·cm$^{-2}$·kyr$^{-1}$, similar to measurements of long-term dust accretion in regional soils[14] (Fig. 5a, e). Interglacial (Holocene, MIS 5, and MIS 7) dust flux is 0.61 g·cm$^{-2}$·kyr$^{-1}$, higher and more variable than glacial dust flux (MIS 2, 3, 4, and 6), which has a median value of 0.38 g·cm$^{-2}$·kyr$^{-1}$ (Fig. 5a). Peak interglacial conditions directly follow glacial-to-interglacial transitions and exhibit the highest dust fluxes, with median values of 0.89 g·cm$^{-2}$·kyr$^{-1}$ during MIS 5e and 1.78 g·cm$^{-2}$·kyr$^{-1}$ during the Holocene (Figs. 4b and 5a). Holocene dust flux values align with observations from the San Juan Mountains of Colorado[3,23,24] and modern measurements throughout the region[25–28] (Fig. 5e). These peak interglacial episodes are 1.2 to 10 times dustier than MIS 2 and 6 glacial maxima.

Fine dust (EM3) constitutes 64% of the total dust signal and drives total dust accumulation trends. It exhibits its highest fluxes and variability during peak interglacials (Holocene and MIS 5e) (Fig. 5b). Coarse dust (EM4) comprises the remaining 36% of the total dust signal. Its flux rate is higher overall and less variable during glacial periods but exhibits higher peak values and variability during peak interglacials (Fig. 5c). Notably, both dust populations contribute to higher total dust fluxes during the MIS 4 glacial (0.7 g·cm$^{-2}$·kyr$^{-1}$) compared to the MIS 2 (0.24 g·cm$^{-2}$·kyr$^{-1}$) and MIS 6 glacials (0.35 g·cm$^{-2}$·kyr$^{-1}$) (Fig. 5a, b, c).

## Discussion

The grain size properties and accumulation rates of sediments in Stoneman Lake indicate that dust is regionally sourced and accumulates more rapidly during interglacial periods. Our end-member modeling approach did not identify a unimodal grain size population <3 μm, a characteristic of far-traveled, intercontinental dust (Fig. 2b). Dust delivered from Asia and Africa must remain suspended for thousands of kilometers as it makes its transoceanic journey, avoiding gravitational fallout and scrubbing from the atmosphere by precipitation. Asian dust, too, must traverse a major orographic boundary that shields all of western North America (Fig. 1a). We instead identify two dust end members with modes in fine and coarse silt, 13 and 38 μm, respectively. Dust accumulation trends at Stoneman Lake are inconsistent with measurements downwind of other major dust sources in the Northern Hemisphere (Fig. 4), suggesting that dust influx at Stoneman Lake is a function of unique regional processes.

EM3 is likely an integration of dust from desert sources to the west and south across the Basin and Range province and accounts for most dust accumulation at Stoneman Lake, driving total dust accumulation trends. EM3's mode (13 μm) is consistent with dust that has traveled in atmospheric suspension hundreds of kilometers from its source[29]. Extensive Quaternary sedimentary deposits including alluvial fan systems, playas, and river floodplains—features principally responsible for regional dust emissions[1,30,31]—are abundant within this range upwind of Stoneman Lake (Fig. 1b). In this area, major dust emissions primarily occur during two annual weather phenomena: westerly cold fronts in spring and convective outflows (haboobs) in summer associated with the southerly North American Monsoon circulation[32–34]. Both patterns generate strong gusts and atmospheric instability that loft dust high into the atmosphere and convey it hundreds of kilometers downwind.

EM3 accumulation correlates to climate and geomorphic activity in the Basin and Range (Fig. 6) and supports a model in which long-term regional dust emissivity and downwind accumulation is principally governed by fine sediment supply on dust-exporting landscape features—alluvial fan systems, playas, and floodplains (Fig. 7). The rate of sediment delivery to these features depends on the erosion and transport of upstream sediment sources, a function of climate, which regulates vegetation distribution, surface runoff, and fluvial

discharge[35,36]. During glacial periods (MIS 2: 34–14 ka; MIS 6: 190–132 ka), EM3 accumulation at Stoneman Lake declines (Fig. 6a), coinciding with a regional increase in effective moisture, i.e., pluvial conditions, resulting from a combination of lower temperatures and steering of excess moisture into the region by atmospheric effects of the Laurentide Ice Sheet[37,38] (Fig. 6b, c). Glacial pluvial climates affected the landscape in ways that would explain the EM3 dust decline. First, playas filled with water[38], suppressing their potential to emit dust (Fig. 7a). Second, as the subtropical jet and associated westerly storm tracks shifted southward into the region, the southerly monsoon circulation and tropical cyclones were displaced[39,40]. Decreasing regional influence of this atmospheric phenomenon may have limited the generation of significant dust emissions by reducing associated gusty winds and extreme precipitation events that generate runoff and cause significant erosion and exposure of fine sediments. Third, glacial climates caused the expansion of piñon-juniper-oak woodlands—currently found at higher elevations—across much of the Basin and Range landscape's lower elevations[41], further stabilizing hillslope soils[36] and limiting the transport of fine sediments into alluvial systems and playas (Fig. 7a). Geomorphic responses in SWNA to pluvial conditions during glacial periods ultimately suppress dust activity and explain the stark difference between this record and most others across the globe, which, in contrast, indicate peak dustiness during glacial maxima coinciding with enhanced glaciogenic dust production (Fig. 4).

Following glacial terminations, early interglacial periods like the Early Holocene (11.7 ka) and MIS 5e (132–116 ka) see dramatic increases in EM3 accumulation (Fig. 6a). Glacial-to-interglacial transitions give rise to rapid warming, increased aridity (Fig. 6b, c), more intense precipitation runoff events[42,43], and an upslope migration of woodland vegetation[44] (Fig. 6d). These changes cause widespread erosion of previously stable, fine sediment-rich hillslope soils[36], mobilizing them into ephemeral washes, alluvial plains, and now-exposed playa beds, where they contribute to regional eolian sediment supplies (Fig. 7b).

After peak interglacial stages, EM3 accumulation trends suggest that shorter orbital-scale wet-to-dry transitions can induce geomorphic responses comparable to full glacial-to-interglacial shifts, albeit with lower magnitude. EM3 flux appears to be influenced by orbital precession (~20 kyr cycles) during the MIS 7 substages a, b, c, and d (243–190 ka) and MIS 5 substages a, b, c, and d (116–72 ka) (Fig. 6a), periods when the climate remained warm and dry but oscillated between slightly warmer-drier and cooler-wetter phases[45–47]. After MIS 5e, EM3 fluxes become notably lower but increase slightly following transitions from cooler-wetter phases (substages b and d) to warmer-drier phases (substages a and c) (Fig. 6b–d). A similar trend is observed in MIS 7, although the record does not extend back far enough to capture the MIS 7e peak.

The Holocene record resolves dust variability on sub-orbital (hundred-to-thousand-year) timescales and clearly demonstrates that drier climates alone do not always correspond to increased dust emissions. While EM3 accumulation remains high throughout the Holocene, our data indicate lower regional emissions during the relatively more arid mid-Holocene (8–4 ka)[48], followed by a marked increase in the late Holocene (4–0 ka) (Fig. 6a). This late Holocene rise in EM3 flux aligns with enhanced ENSO variability[49], which is linked to alluvial fan aggradation and increased flooding across SWNA[50–52]. These findings underscore the potentially critical role of precipitation runoff and erosion in providing sediment to the region's eolian system, reinforcing studies that link dust emissions to hydrologic variability via geomorphic activity and sediment supply rather than directly to aridity alone[3,4,53].

EM4 accounts for the remaining dust accumulation at Stoneman Lake and, given its much larger modal grain size (38 μm), likely signals more proximal dust emission processes than EM3. EM4's mode is consistent with dust that has traveled only tens, rather than hundreds, of kilometers[29]. Much of EM4 likely originates from the Verde Valley, a

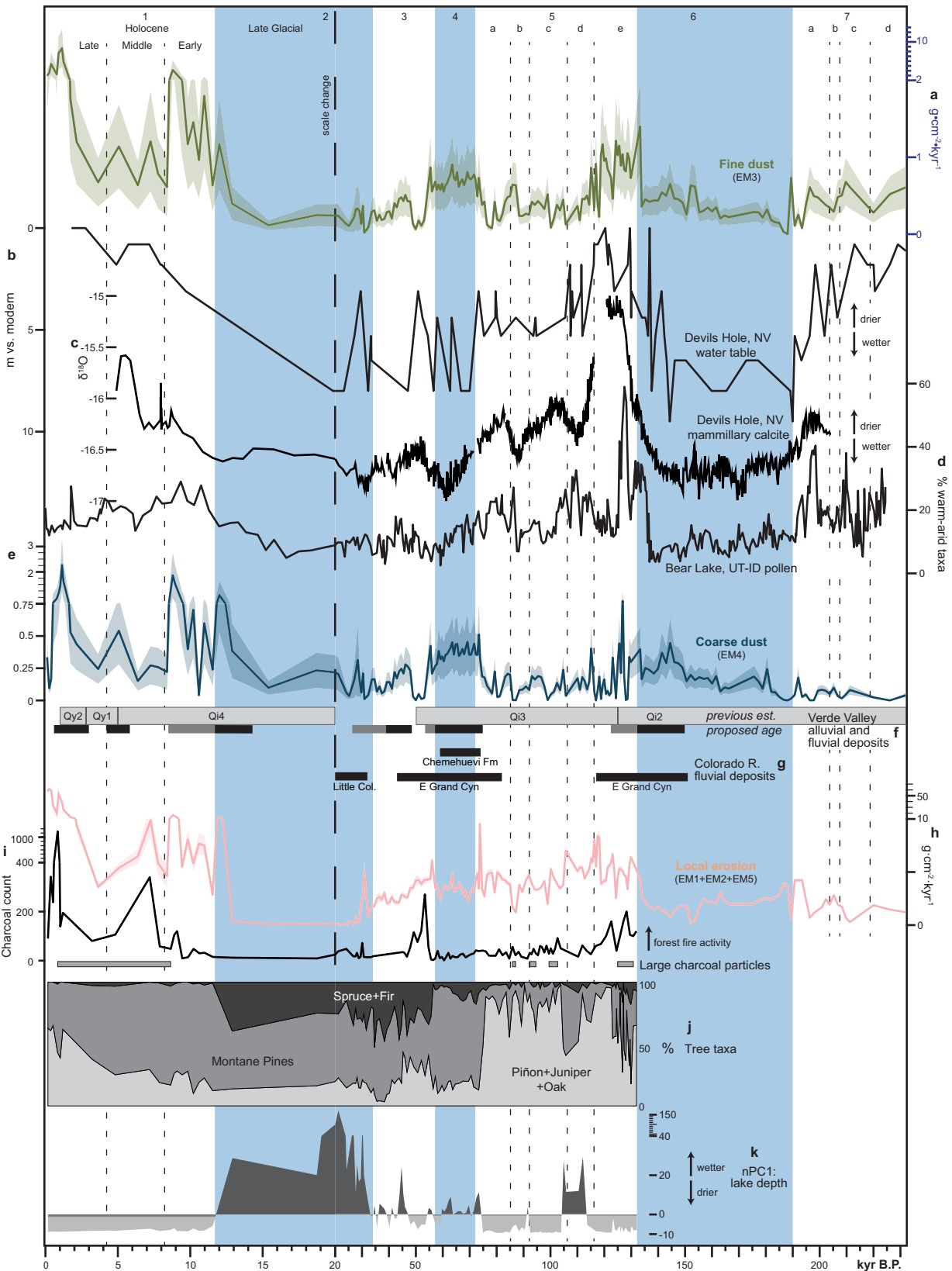

-50-km long half-graben with a broad, gently sloping valley floor at ~1000 m elevation (Fig. 1c) representing the Basin and Range feature closest to Stoneman Lake. Quaternary deposits of the through-flowing Verde River, its tributaries, and flanking alluvial piedmonts contain abundant fine-grained sediments and are located just 25–45 km to the W–SW[54–60] (Fig. 1c). These deposits represent the largest and most

proximal area to have likely contributed considerable dust to Stoneman Lake.

Sustained increases in EM4 accumulation rate correlate with periods of fluvial aggradation in the Verde Valley. On average, EM4 accumulates more rapidly during glacial periods (Figs. 5c and 6e), corresponding to periods of fluvial deposition along the Verde River[54–60]

**Fig. 6 | Climate, geomorphic activity, and clastic sediment fluxes into Stoneman Lake.** Note the change in the x-axis scale at 20 ka. Glacial-interglacial cycles numbered by Marine Isotope Stage (MIS) with strong glacials shaded blue and substages indicated by dashed lines. Substage boundaries during MIS 5 generally align with 20-kyr precession cycles. **a** Fine dust mass accumulation rate at Stoneman Lake, indicating emissions occurring 100's of kilometers upwind across the Basin and Range province. **b–d** Indicators of hydroclimate in the Basin and Range province. **b** Water table height (inverted) at Devils Hole, Nevada[67]. **c** $\delta^{18}O$ of cave wall calcite (inverted) at Devils Hole, Nevada[46]. **d** Percentages of warm-arid taxa from Bear Lake, Utah-Idaho[47]. **e** Coarse dust mass accumulation rate at Stoneman Lake, indicating emissions occurring 10's of kilometers upwind in the Verde Valley. **f** Age ranges of fluvial and alluvial deposits in the Verde Valley[54–60]

(light gray, labeled by map unit name). New interpreted age ranges of fluvial (black) and alluvial (gray) units based on correlation to (**e**). **g** Fluvial aggradation episodes in the Colorado River system, including lower Colorado River deposits of the Chemehuevi Formation[62], the Little Colorado River[53], and in the Eastern Grand Canyon[61]. **h** Flux of local clastic material into Stoneman Lake representing trends across the northern and eastern uplands of the Verde Valley. **i–k** Climatic and paleoecologic indicators from STL14 core[63]. **i** Charcoal count indicating wildfire activity (large particles (60–100 μm) indicated by gray bars). **j** Tree taxa percentages reflecting local forest composition. **k** Principal component analysis of non-pollen palynomorphs. Higher nPC1 scores indicate oligotrophic conditions and a higher lake level. Error envelope of mass accumulation rates (**a**, **e**, **h**) based on percent error propagation of values factored into their calculation.

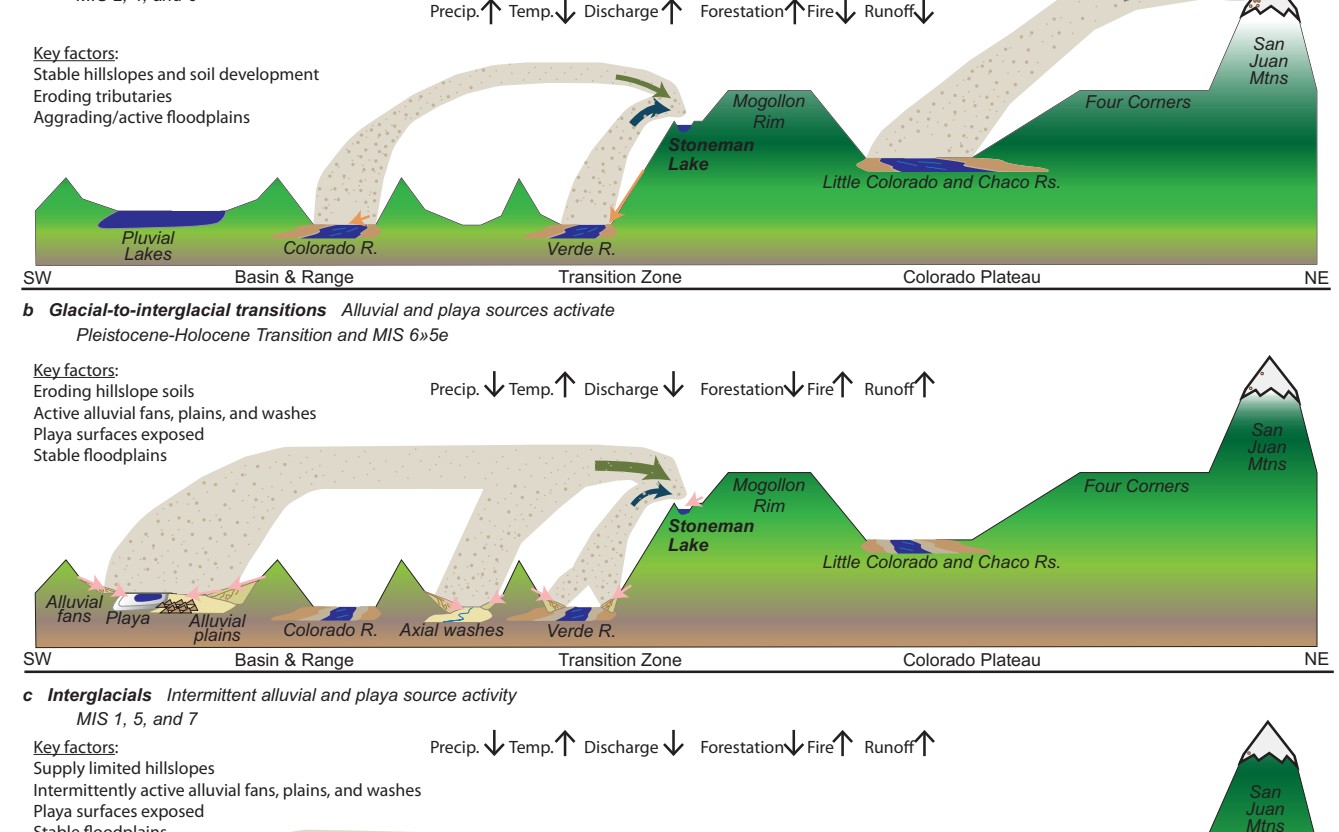

**Fig. 7 | Climate, geomorphic activity, and eolian sediment transport in SW North America. a** Hillslopes are generally stable during glacial periods, limiting fine sediment delivery to and exposure on alluvial systems. Pluvial lakes limit playa emissions. High baseflow conditions erode fluvial tributaries and deliver sediment to mainstem floodplains, where they contribute to dust emission. (MIS = Marine Isotope Stage). **b** Glacial-to-interglacial transitions lead to widespread geomorphic instability on hillslopes. Hillslope sediments are transported downstream into alluvial systems and terminal basins, where they become exposed, promoting dust emission. **c** Following this initial pulse of sediment, interglacial periods are marked by more supply-limited conditions on alluvial and playa dust sources. Low baseflow conditions confine products of hillslope erosion to tributaries, limiting fluvial dust source emissions.

(Fig. 6f). As these deposits aggraded the Verde River floodplain, abundant fine-grained sediment would have been exposed to the atmosphere (Fig. 7a). Similar conditions of increased fluvial sediment load and floodplain aggradation during recent glacial periods are observed across many reaches of the middle and lower Colorado River system as pluvial

conditions increased fluvial discharge, transporting tributary sediments into mainstem river corridors[53,61,62]. Conspicuously departing from the prevailing trend of dustier interglacials, the sustained increase in EM3 accumulation during glacial MIS 4 (72–57 ka) (Fig. 6a) correlates to one of the region's most extensive fluvial aggradation events, during which

the entire lower Colorado River valley along the Nevada-Arizona-California border filled with >100 m of mostly fine-grained sediment, forming what is now the Chemehuevi Formation[62] (Figs. 1a, 6g, and 7a). This sustained and spatially extensive aggradation would have provided a consistent source of regional dust throughout MIS 4. Thus, while regional fluvial source emissions appear out of phase with those from desert alluvial and playa sources, dust from large fluvial aggradations may exert a similar influence downwind.

While EM4 accumulation is higher during glacials on average, its variability increases in magnitude during early interglacials, suggesting a shift from fluvial to alluvial dust source emissions in the Verde Valley (Figs. 5c and 6e). Compared to more prolonged episodes of floodplain aggradation, alluvial fan aggradation tends to occur more rapidly and with higher frequency. Following glacial-to-interglacial transitions, drier conditions overall likely lead to lower discharge and sediment flux into the Verde River, restricting significant floodplain aggradation and associated dust emissions. At the same time—much like across the rest of the Basin and Range—upslope migration of woodland vegetation and flashier precipitation runoff events likely triggered hillslope soil erosion (see below), forming the extensive axial alluvial deposits in the Verde Valley[54–60] and periodically exposing fresh fine sediments to the atmosphere. If EM4 accumulation is indeed a direct signal of fluvial and alluvial aggradation in the Verde Valley, periods of increased accumulation may constrain their depositional ages (Fig. 6f), currently approximated based on soil morphological development.

Because Stoneman Lake is situated in the higher elevations of the Verde Valley watershed (Fig. 1c), local erosion rates (EM1 + EM2 + EM5), in tandem with local paleoenvironmental records[63], demonstrate how upland processes eventually contribute sediments to nearby dust sources downstream in the Verde Valley. Reduced upland erosion rates during glacial periods, especially during glacial maxima of MIS 2 and MIS 6 (Fig. 6h), correspond to increases in effective moisture, expansion of montane and subalpine forests to lower elevations, and reduced wildfire activity (Fig. 6i–k). Transport-limited conditions on upland hillslopes likely promoted widespread soil development (Fig. 7a). Despite minimal sediment export from the upland environment, evidence of nearby floodplain aggradation suggests significant quantities of sediment were introduced to the Verde River mainstem (Fig. 6e, f), presumably from intervening tributary environments (Fig. 1c).

Glacial-to-interglacial transitions at 132 ka (MIS 6 > 5) and 11.7 ka (the Pleistocene-Holocene Transition) introduced rapid changes throughout the Verde Valley watershed that destabilized the land surface and rapidly accelerated the erosion of hillslope soils amassed during the preceding glacial period (Figs. 6h and 7b). These changes include rising temperatures, reduced canopy cover at lower elevations (Fig. 6j), increased wildfire activity (Fig. 6i), and more intense, though less frequent, rainfall events producing flashier runoff. Reduced tributary baseflow and enhanced hillslope erosion during the Holocene and MIS 5e align with the shift in local dust source from floodplains to axial alluvial fans perceived from EM4 accumulation rate variability (Fig. 6e).

Sediment transport processes following peak interglacial conditions are captured during MIS 5. After MIS 5e, upland erosion rates remain elevated, decreasing only slightly (Fig. 6h), correlating to continued open forest canopies, wildfire activity, and drier conditions (Fig. 6i–k). In contrast, low rates of EM4 accumulation indicate limited sediment transport into the lower elevations of the Verde Valley (Fig. 6e), likely reflecting a stable or incising mainstem floodplain environment and depletion of hillslope sediment stores necessary for alluvial fan aggradation (Fig. 7b). Thus, eroded upland sediments likely accumulated in tributary environments during interglacials, only to be flushed into the mainstem as baseflow conditions increased during subsequent glacial periods (Fig. 7a). These observations highlight the episodic nature of sediment mobilization and the important role of glacial-to-interglacial transitions in delivering sediment from hillslopes to fluvial and alluvial systems in this semi-arid environment.

Although trends in dust accumulation at Stoneman Lake are well explained by regional climate and geomorphic responses, it is worth considering whether accumulation rates could instead be controlled by eolian processes that follow emission, i.e., transport and deposition. Dust from southern sources in the Sonoran Desert has a slightly shorter pathway to Stoneman Lake (180–400 km) than western source regions in the Mojave and southern Great Basin deserts (250–600 km, Fig. 1b). Coarser EM4 might then reflect the intensity of the southerly monsoonal pathway and EM3 the westerly pathway driven by cold front passage[4]. The shifting influence of these synoptic-scale circulation patterns in SWNA on glacial-interglacial timescales allows us to test this interpretation. During glacials, westerly storm tracks shift southward into SWNA, displacing the southerly monsoon circulation[39]. After deglaciation, westerlies shift back northward, allowing the monsoon to re-enter the region. If the relative influence of these atmospheric pathways is the dominant control on dust accumulation rates at Stoneman Lake, EM3 would accumulate more rapidly during glacials than interglacials, and EM4 should accumulate more rapidly during interglacials than glacials. Instead, our data show the opposite pattern (Figs. 5b, c and 6a, e), indicating that changes to the region's dominant synoptic circulations and dust transport pathways are likely not the primary control on regional dust flux.

We also find minimal evidence that precipitation-induced fallout and surface roughness influence long-term dust deposition rates. Dust fallout is enhanced during precipitation events as dust particles form the nuclei for raindrops. However, our data does not show increases in total accumulation during wetter glacial periods (Fig. 5a). Increasing surface roughness from denser forestation at Stoneman Lake during glacials[63] would theoretically enhance dust deposition by slowing surface winds, yet our observations reveal no such correlation. With little evidence of primary influence by mechanisms of transport and deposition, we maintain our original interpretation that dust accumulation rates at Stoneman Lake are predominantly controlled by the emissivity of regional dust sources.

It is also important to consider if the Stoneman Lake record is representative of the broader region. While top-down glacial-interglacial climate changes are felt similarly across SWNA, the geographic distribution of fluvial versus alluvial and playa dust sources may result in unique, location-dependent patterns of dust accumulation. For instance, the Stoneman Lake dust accumulation record is out of phase with a comparable record from the San Juan Mountains of Colorado[3] (Figs. 1b and 8a, c). Unlike the Stoneman Lake record, dust deposition in the San Juans is much higher during the Late Glacial, declining sharply after the Pleistocene-Holocene Transition (11.7 ka) and remaining low throughout the Holocene (Fig. 8). Several fluvial systems upwind of the San Juans, including the Little Colorado and Chaco Rivers (Fig. 1b and 7), were active during the Late Glacial and acted as sources for proximal dune construction until about 12 ka[53,64,65] (Fig. 8b). Fine-grained material from these fluvial sources was likely delivered downwind to the San Juan Mountains—but not upwind to Stoneman Lake. In contrast, Stoneman Lake likely receives most of its dust from alluvial and playa sources in the Basin and Range, which activated after the Pleistocene-Holocene Transition (Fig. 8c). The divergence between the two records may therefore reflect the spatial distribution of their primary dust sources relative to prevailing winds and the contrasting responses of fluvial and alluvial dust sources towards the same regional hydroclimatic change.

Dust flux variability in SWNA appears to be primarily governed by the supply of wind-available fine sediment across upwind dust sources rather than direct climatic drivers such as aridity. Dust supply is primarily determined by the climate-paced storage and release of sediments from regional hillslopes and fluvial tributaries. On hillslopes, sediment stores are developed during glacial periods and then rapidly released upon deglaciation, activating downstream dust sources. In tributaries, sediment accumulates during interglacials and is released

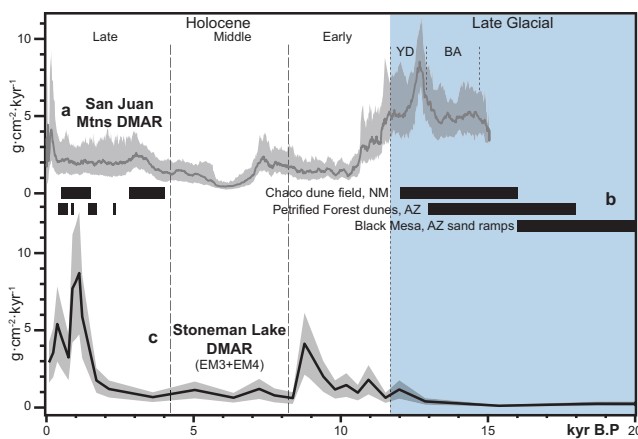

**Fig. 8 | Spatiotemporal patterns of dust accumulation on the Colorado Plateau.**
**a** Dust mass accumulation rate (DMAR) in the San Juan Mountains showing higher flux prior to the Pleistocene-Holocene Transition[3] (YD = Younger Dryas; BA = Bølling-Allerød) correlating to **b** Dune construction events along the Little Colorado and Chaco Rivers during the Late Glacial[53,64,65]. Prevailing winds situate these fluvial dust sources to deliver dust to the San Juans but not Stoneman Lake (Fig. 1b). **c** Total dust mass accumulation rate at Stoneman Lake indicates increased dust accumulation following the Pleistocene-Holocene Transition, highlighting the primary influence of alluvial and playa dust sources in the Basin and Range province.

during subsequent glacial periods as fluvial discharge increases, aggrading mainstem floodplain environments downstream. Thus, this dust accumulation record builds our understanding of the history of alluvial and fluvial processes that continually reshape the landscape and, in the process, erode other direct evidence of their activity. Given the dominance of alluvial and playa dust sources across the Basin and Range province and their particular responses to glacial-interglacial change, areas downwind are distinctly dustier during interglacial periods than glacial periods.

## Methods

### Age model
Core STL14, recovered in 2014, preserves a sediment archive of the last ~1.3 million years[45]. The core's current age model[18] (Supplementary Fig. 1 and Supplementary Data) integrates radiocarbon dating[45,66], tephrochronology[45], and climostratigraphy[18,45]. The climostratigraphic approach is supported by correspondence of various stratigraphic datasets[18,45,63] to global climate changes at Marine Isotope Stage (MIS) and MIS substage scales as well as changes observed in other regional records with independent age control (e.g., refs. [46,47,67]). The age model was constructed using *R*-based Bayesian age modeling software, Bacon v.2.3[68]. We use the median age model for our analyses and visualizations.

### Sampling, laser granulometry, and end-member modeling analysis
Volumetric sampling of core STL14 was conducted at Northern Arizona University's Laboratory of Paleoecology, where the working half of the core is stored (the archive half is stored at CDSCO at the U. of Minnesota). The median sample resolution by climate stage is as follows: Holocene: 0.4-kyr; Last Glacial: 0.7-kyr; MIS 5: 1.0-kyr; MIS 6: 1.6-kyr; MIS 7: 1.5-kyr. The clastic fraction of each sample was isolated through successive chemical washes of 30% $H_2O_2$, 1 N HCl, and 1 N NaOH to remove organic matter, carbonate, and biogenic silica, respectively. Previous quantitative X-ray Diffraction (qXRD) and sedimentological analyses indicate negligible carbonate content in sediments younger than 160 ka, with diagenetic concentrations in older sediments[18] that

were avoided during sampling. Processed samples were then sent to the Paleoclimatology Laboratory at the California State University, Fullerton, for laser granulometry using a Malvern Mastersizer 2000 coupled to a Hydro 2000 G dispersion unit. Silica carbide standards were analyzed twice at the start of each run, once every ten samples thereafter, and once at the end of each day of analyses to assess equipment stability, reproducibility, and analytical error. Since 2017, a total of 4129 standards have been analyzed, averaging 13.11 microns (μm) with a standard deviation of 0.10 μm.

Particle size distributions (PSDs) from all 205 core sediment samples were processed using AnalySize software[69], an end-member analysis MATLAB GUI package designed for unmixing PSDs into constituent end-member (EM) populations. We employed a non-parametric approach to identify multimodal populations, as expected for locally sourced sediments. AnalySize runs multiple scenarios that fit 1–10 end members to the measured grain size dataset, returning model performance statistics alongside PSDs and percent composition for each modeled end member (Supplementary Fig. 2). Scenarios with 3, 4, and 5 end members (3EM, 4EM, and 5EM) were selected for further consideration based on their fit to the measured data, with total model fit ($r^2$) values of 96%, 98%, and 99%, respectively. These scenarios also exhibited minor intercorrelation among constituent end members, suggesting avoidance of model overfitting (Supplementary Fig. 2a, b).

Similarities in modeling results for EM1 across each scenario (3EM, 4EM, and 5EM) suggest a robust model result for EM1 (Supplementary Fig. 2c–e). Each scenario's EM1 has a primary mode at 0.18 μm and one to two secondary modes in the clay to silt range, indicating poorly sorted, i.e., locally sourced, sediments. Each model scenario produced similar stratigraphic abundances of EM1 (Supplementary Fig. 3).

End members 3EM3, 4EM4, and 5EM4 show similar unimodal distributions (31, 42, and 38 μm) and stratigraphic variation (Supplementary Figs. 2c–e and 3), consistent with short-range-transport dust (sourced from <100 km away) and coarse dust particle size distributions identified in other regional studies[3,4,22]. Here again, similarities between each scenario suggest a robust model result.

The primary distinction between the scenarios lies in the modeling of sediment populations in the very fine to fine silt range. In the 3EM scenario, 3EM2 exhibits a unimodal peak at 9.5 μm (Supplementary Fig. 2c), characteristic of medium-range-transport dust (sourced from 100s of km away). If both 3EM2 and 3EM3 are interpreted as dust, this scenario suggests no input of local alluvium during interglacials MIS 7 or MIS 5e (Supplementary Fig. 2c), contradicting geochemical and palynological evidence indicating vigorous sediment transport during these periods[18,63]. The 4EM scenario appears to divide 3EM2 into finer (4EM2: 6 μm) and coarser (4EM3: 13 μm) end members (Supplementary Figs. 2d and 3). We interpret 4EM2 as resuspended fine sediment from the lake margin (e.g., ref. [3]). This process occurs during early interglacials when lake levels drop, exposing lake bottom sediments to erosion, and accounts for high accumulation rates of 4EM2 during MIS 7, 5e, and the Holocene (Supplementary Fig. 3).

Ultimately, we selected the 5EM scenario, as it retains the EMs of the 4EM scenario while introducing a fifth end member (primary mode at 169 μm; secondary modes at 0.18 and 19 μm) that enhances model fit during specific time periods around 31 and 12 ka (Supplementary Fig. 2e) when coarser material intercalated the lake stratigraphy. 5EM5 appears to be another locally sourced sediment population, given its coarse and multimodal PSD. In summary, the 5EM scenario maximizes the variance accounted for by the end-member model (Supplementary Fig. 2a, b), reduces end-member intercorrelation (Supplementary Figs. 2a, b and 4), and yields end members whose PSDs align with those from other regional dust studies. Furthermore, accumulation trends and source interpretations of end members in the 5EM scenario are consistent with geochemical measurements by Staley et al.[18] (see Text).

## Sediment flux calculations

The mass accumulation rate (MAR) of a sediment population is calculated by multiplying the sedimentation rate by the fraction of that population in the bulk sediment (determined by the end-member model) and its density[17]. Because this study focuses on clastic sediment MARs, the fractional volume of non-clastic material, including organic matter, biosilica (bSi), and authigenic minerals, and its effect on dry bulk density (DBD) must be determined.

Non-clastic material was determined during previous studies (Supplementary Fig. 5). Total organic carbon (TOC) was measured at a median sample resolution of 0.74-kyr[45]. Modal percentage estimates of bSi (diatoms and sponge spicules) were derived from smear slides at a median sample resolution of 2.8-kyr[45]. Authigenic carbonate content was quantified from smear slide observations[45] and qXRD[18].

Using DBD as a proxy for clastic dry density ($DD_{clastics}$) is complicated by the presence of low-density organics and bSi. To address this challenge, we applied linear regression to DBD below 9.05 meters composite below lake floor (mcblf)—where TOC and bSi are negligible, resulting in DBD ≈ $DD_{clastics}$—to adjust DBD values in the upper portion of the core (0–9.05 mcblf) where TOC and bSi are nonnegligible. We first normalized the DBD values above 9.05 mcblf to the linear regression line of the DBD measurements below 9.05 mcblf (Supplementary Fig. 6). Next, we constrained the normalized curve to maintain (1) continuity with the DBD curve at 9.05 mcblf and (2) consistency with the original linear regression, ensuring that both curves share the same linear best-fit equation. The adjusted curve, $DD_{clastics}$, lies primarily within 1 standard deviation (σ) of the linear best-fit line. Notably, below 9.05 mcblf, −1σ excursions of DBD are coincident with brief increases in bSi (Supplementary Fig. 6). We assume that the dry densities of each modeled EM are equivalent to $DD_{clastics}$.

Due to uneven sample spacing of all factors involved in MAR calculations, each dataset is linearly interpolated to the resolution of the particle size dataset. The MAR of clastic end member x ($MAR_{EMx}$) is calculated as:

$$MAR_{EMx} = SR * f_{EMx} * DD_{clastics} \qquad (1)$$

where $SR$ is the sedimentation rate derived from the median age model, $f_{EMx}$ is the bulk fractional composition of $EM_x$ relative to other clastic end members and non-clastic content (TOC, bSi, and TIC), and $DD_{clastics}$ is the clastic dry density (Supplementary Fig. 5). MAR calculations for dust end members (EM3 and EM4) are multiplied by a lake-to-catchment area scaling factor (0.63/3.54) to account for catchment focusing effects[24]. Total dust mass accumulation rate ($DMAR$) is the sum of $MAR_{EM3}$ and $MAR_{EM4}$, while MAR of local clastics ($MAR_{LC}$) is the sum of $MAR_{EM1}$, $MAR_{EM2}$, and $MAR_{EM5}$

Denudation rate of catchment basalt ($DR$) is derived directly from $MAR_{LC}$ by dividing by the density of basalt and multiplying by a catchment area scaling factor that assumes no erosion from the lake floor:

$$DR = \frac{MAR_{LC}}{p_{basalt}} * \frac{Lake\ area}{Catchment\ area - Lake\ area} \qquad (2)$$

This calculation relies on the assumptions that (1) material is detached from bedrock and transported into the lake immediately, which may not hold true during periods of geomorphic stability (e.g., MIS 2 and MIS 6a) and (2) no mass is lost to solution in through-flowing groundwater. To address these issues, we report $DR$ as an integrated value representing average denudation of the catchment since ~230 ka and treat it as a minimum value.

Analytical error for each calculation is assessed using a percent error propagation method that incorporates errors in bulk composition, density, and catchment area scaling factors. Inputs for our uncertainty calculations include a 3% error for density to account for

potential water retention after drying the samples[70], and a 4% error for TOC, as reported by UNM's Center for Stable Isotopes. Given the inherent uncertainties in estimating modal abundances of sediment constituents from smear slides, we apply a conservative 20% error for the sum of authigenic carbonate and bSi. The most significant source of error arises from estimating the area of lake floor deposition. The current lake floor covers approximately 0.63 km². We establish an upper limit of 0.86 km² that corresponds to the area enclosed by the bedrock slope-to-fan apron transition—a distinct topographic break surrounding the lake (Fig. 1d). This slope transition represents a definitive boundary above which sediment accumulation is unlikely. Conversely, we define a lower limit of 0.38 km² based on the low-stand lake area of the most recent megadrought, which roughly corresponds to the area currently occupied by reeds, the darker area in the middle of the lake. This range of uncertainty corresponds to a 40% error in the lake-to-catchment area adjustment factors. It is worth noting that changes in lake depth likely influenced the size of the depositional area on the lake floor, increasing during glacial periods and decreasing during interglacials. Therefore, glacial MAR values may fall within the upper uncertainty envelope, while interglacial MAR values may reside within the lower envelope. Therefore, we report the factor by which DMAR changes between glacial maxima and peak interglacial conditions as a range that incorporates all the above uncertainties.

## Data availability

The source data generated and analyzed in this study are provided in the Supplementary Information.

## Code availability

Computer code to perform end-member modeling analysis in AnalySize[69] is available from (https://github.com/greigpaterson/AnalySize/releases).

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

## Acknowledgements

The authors wish to acknowledge assistance from the US Forest Service during core collection, the Pilot Grant Program (PGP) grant from NAU for funding core recovery, a Research Allocation (RAC) Grant from UNM, and a New Mexico Research Grant (NMRG) from the UNM Graduate and Professional Student Association. We also thank the core research support provided by the Continental Scientific Drilling Facility, University of Minnesota.

## Author contributions

S.S. designed and performed data analyses, wrote the manuscript, and prepared figures. P.F., R.S.A., and S.S. acquired funding for the research. P.F. supervised this project. M.K. performed laser particle size analysis. P.F., R.S.A., and M.K. contributed to the manuscript.

## Competing interests

The authors declare no competing interests.
