## [Transparent Peer Review file · Nature Communications]

Higher interglacial dust fluxes relative to glacial periods in southwestern North American deserts

Corresponding Author: Dr Spencer Staley

Version 0:

Reviewer comments:

Reviewer #1

(Remarks to the Author)

Dear Editor,

Herewith my review of Staley et al. submitted to your journal.

The authors report on a very comprehensive of an interesting lacustrine record across the last 200 kyr of time. They focus on dust accumulation over time and whether or not changes occurred during glacial-interglacial cyclicality. Such a study could be of interest for Nature Communications searching for studies with a broader geoscientific audience and impact. Here, are my main thoughts and concerns with the study. Below, you find Line-by-line thoughts:

- the study has not been uniquely sent to your journal. The dataset seems to be completely sent to Geology also, where another part of the study is being discussed. This makes the current work less attractive, as the work is split into two, but also less holistic. Why would one discuss the local dust transport separate from the regional dust transport? In the end, it is the same region and one cannot be seen separate from the other? The authors wish to publish in two of the better journals in their fields, why not combine that into one great study rather than two intermediate studies?
- There is not a proper uncertainty propagation when the differences are calculated (standard errors) of dust fluxes between climatic stages. The 1.6 times change in flux should be going along with a standard error to know what is its significance. This nor any of the other 'differences' nor standard errors are quantified in the submitted paper. The key figure providing fluxes per stage is rather small and not sufficiently explanatory (see below minor comments)
- The Results section jumps into interpretations from the second line, while many results are not described and only come in the Discussion section. The Discussion section has a too high degree of certainty given in words to interpretations. As most interpretations come back in the Discussion anyway, it is needed to clean the Results section from interpretations and appropriately describe interpretations as possibilities in the Discussion section.
- The Discussion section dwells upon details of functioning how it all may have happened, the model, while I miss a much more straightforward line of reasoning from data to simple interpretations. Some claims further in the Discussion section are more 'review paper' like than that these relate directly to data in the current paper, also because the 'other half' of the data are not discussed here. The authors may have gone much further in their reasoning and describe that, while forgetting that the reader has to do it with what is currently provided. A more direct and concised Discussion chapter will help the paper.
- For a wider audience, it is needed to write with much less abbreviations and with less assumptions as if the reader would know things, see minor comments. The reader for Nature Communications should be regarded as an interested geoscientist, nothing less and nothing more. Different writing is needed for that.

Based upon the above thoughts, I recommend to ask the authors to combine this study with that submitted to Geology into one, very solid study with high impact. Besides, the current work needs a major writing revision, while the science seems appropriate.

Minor Comments / Ideas / Suggestions:

Line 16: it is needed to add the standard error of the difference to this number, such to see the relevance of this quantification within the total uncertainty of the data.

Line 73: The second sentence of the Results section brings an interpretation. It would be better to first describe the results.

Line 71 Results: please start the results section with a basic description of the main results, namely Laser Granulometric results of the samples.

Lines 81-85: EM-2 would be perfectly in line with long-range dust source, the reasoning here seems a bit weak?

Line 104-105 Ruling far distance dust transport is a bit dangerous, I would think. First, what about relating EM-2 to long

distance transport?

Line 111: The authors should use less abbreviations in the running text: DMAR, MRT, SRT, etc. and in particular not without explanation in each paragraph. The journal is aimed at a wider audience than specialists. This whole paragraphs reads great for those involved in the study, but is impossible to read for a (relative) outsider. This could also include finding other words for these items, MRT does not even include the 'dust'.

Lines 122-125: here it is needed to present all results: Figure 4 is only partially presented and the key number from the abstract including its uncertainty does not figure in the Results.

Line 127: The phrasing is not scientific. EM-3 is an end-member peak in a end-member modelling scenario representing the 13 micron grain size mode. EM-3 could relate to Medium-range transport dust, that, if that would be the case, is likely of North-American origin. It is an interpretation, not a fact. The authors have to rewrite the text making clear distinction between facts (a measurement, 13 micron mode) and an interpretation (MRT, North American source).

Line 132: write in full STL

Lines 143-145: please explain "cannot be controlled by intervening processes of transport and deposition". Who controls what? What is intervening what? Is the MRT interpretation impossible to validate?

Line 155: it would help if the authors write more to a general audience also by more explicit explanations. Here it is assumed that people know what is 'wet removal', that the glacials have the highest effective moisture.

Lines 154-156: isn't this partially a circular reasoning? First, it is assumed that a certain grain size class is purely dust, and then, because there is higher dust flux, there is not much wet removal?

Line 159 SWNA?

Lines 159-160: If MRT is renamed to desert dust, why not initially starting with that name, in my view, better than MRT as more explanatory, and better than using two names in the running text.

Line 159: Evidence 'dictates', not sure if that is really the appropriate academic wording.

Line 162: I propose to include the grainsize mode also in between brackets, SRT, EM-4, 38 micron). Particularly the grainsize is what helps the reader to memorize the data and grains, EM-4 is different for every study, as is SRT.

Lines 163-164: this should be in the Results. And, looking at Figure 4, there is no mention of SRT, so where should I look?

Lines 166-167 Please explain "whose published depositional ages correlate to MARsrt increases".

Lines 172-173: meaning the same record is being published elsewhere? One comprehensive study would have been nicer as SRT and MRT are not that different?

Line 177: SWNA in the title is not very explanatory to a wider audience

Lines 208-209: I do not see this in Figure 3: the transition from MIS 6 to 5 is rapid in the Desert Dust curve given, where is the 20kyr lag?

Lines 232-233: Provide the reader means to find this in the figure, you mean to look at c, d, and e? Relating the tiny spike at 7>6 transition seems too far, the overall agreement in trends between b and c/d/e is, however, worthwhile more reference.

Line 258: The lack of a sustained vegetation cover in glacial times is a primary reason for increased dust fluxes, or do the authors include that in the 'dust production'?

Lines 260-263: that seems a very straightforward conclusion that one may derive from the data, why was it not brought forward earlier? In the relatively complex discussion before this part, there was very little discussion about vegetation.

Lines 260-263: there must vegetation records for comparison to corroborate this claim?

Lines 266-267: Figure 5 gives a clear lower STL MARsrt for the mid-Holocene compared to Early Holocene and Late Holocene, that is not suffice?

Lines 269-274: higher effective moisture means more vegetation cover and less exposure and so lower dust transport. I do not see why the current data are adding to the specific discussion of climatic perturbations other than that reasoning. It seems the discussion is rather complex for a not such a difficult case?

Figure 2: why are the real data hidden behind the end-member analyses? Choices for line thickness are such that the real data are not easy to trace in the figure.

Figure 4: I find this figure / table hard to read, it could be my age, but it all seem rather small, particularly because the red outliers need so much space that the averages and 2.7sigma intervals are somewhere near the bottom of each panel. Would it be an option to leave the outliers for a figure in the appendix and blow up this panel focussing on the main box and whisker plots?

Figure 4: I cannot find the paper nor Methods section what intervals are included in 'Glacial' and what in 'Interglacial'. Same for Early, Middle, Late, add a description in the text / caption. It would help to indicate early, middle, late also to Figure 3 on the time axis. While dividing into glacial / interglacial seems evident, it may not necessarily be. One could argue to only compare Stage 1 and 5e with Stages 2,3, and 6. Or more indendant, a 'cutoff' could be used on the LR04 isotope curve to decide which data point bins in what stage, glacial or interglacial. It may make the results more significant and definitely more independent.

Figure 4: labels 'e', 'abcd' are unclear, these all belong to MIS5? It seems there is space on the image to show that.

Reviewer #2

(Remarks to the Author)

This is a solid study of the dust flux in North America southwest, adding to a growing body of knowledge of the relationship between dust production and climate. The noteworthy results include 1) identifying dust in lake deposits; 2) displaying high dust flux in interglacials than glacials; 3) trying to clarifying the relationship between dust production and climate. The study is valuable, but I think the discovery is incremental and the evidence of some interpretations (particularly the basis of 5 end members and why they have different sources and transported by different wind) is vague. See detailed comments below

Introduction: I do not see the tight connection of the research topic and justification of scientific importance in introduction. Ice cores and deep sea are great proxies for global dust budget. The importance of American southwest to the global picture is

quite vague here, and there is no critical knowledge gap or debates described to clarify the contribution of this study. The human impact on dust generation also seems not relevant given the study extends back to the late Pleistocene. I see the importance at the end of the discussion, but here it is not appropriately sold.

Background: the text lacks key description of the background (description of figure 1), particularly the climate and atmospheric circulation patterns.

Results: 1) why choose five end members? Is this based on observation and sound geologic interpretation or just math technique? I think it is the latter, which, in my opinion, does not have geologic significance. I understand there is knowledge of modern dust source (like Tau et al., geology) that this study is based on, but the reasoning is not well clarified. 2) The explanation of each end member seems very casual. For example, how physical and chemical weathering affect the minerals ton have EM1 and EM5? I know the reference is cited, but such interpretation is the core of the study and should be clarified here. 3) if EM 1 coalesces with EM2, what is the basis to separate them?

How does the geochemical data support? It needs some clarification.

Discussion:

Section 1: 1) the evidence to rule out "processes of transport and deposition" is not clear; 2) Westerlies may be stronger during glacier maximum, what about monsoon? The evidence ruling out wind intensity change is also not compelling. 3) cite reference and evidence for that the highest effective moisture was during glaciers. 4) evidence for the different source of SRT and MRT?

5) Anderson et al say that "fluvial aggradation beginning during full-glacial times and continuing into subsequent interglacial", is this consistent with the observation of SRT?

Section 2: line 177-179, this observation is limited to mid-latitude, right? How about desert, why that is not included given desert to the north and northwest of China loess plateau has been suggested to be a source. Line 181, the transition from alluvial source to desert is very confusing. Line 182, if I understand right, this is for MRT, why not replace "fine" with the grain size for clarity. Any evidence of production of this grain size range at desert hillslopes? line 194 and 197, it is not clear what are the climate-driven surface processes, and how they were controlled by climate and ecological changes. Line 209, I thought with the stage set up earlier in this paragraph, here you want to talk about the slope instability during the interglacial, not sure why you change to talk fluvial adjustment to sediment load increase?

Line 261-263, this seems different from what has been said earlier. Is this a comparison of glacier maxima to the rest of the glacier periods? I think the information provided so far is that glacier is less dustier than interglacial.

I think the rest of the discussion is good and interesting, but they are based on discussion in the first two sections. The lack of evidence and clarity in the first two sections undermines the extended discussion.

Reviewer #3

(Remarks to the Author)

This is an interesting paper turning the heads on peak dust flux during an interglacial, rather than the wet glacial in the SW USA. The basis of this inference is the sediment record from Stoneman Lake, a small lake in central Arizona. This lake has been well studied and dated with a high-quality paper in GSAB. This is a new and important record to sort out hydrologic system response to late Quaternary climate variability.

The central inconsistency of this paper is the lack of granulometry data from the surrounding small catchments to identify local water-transported flux of particles versus aeolian dust flux. There needs to be a detailed study of the source to sink flux of particles from numerous sources around the lake including catchments, overland flow, and aeolian proximal and distal sources. What is the geomorphology and soils around the lake? Vesicular A horizons, a common desert soil morphology, are loaded with aeolian particles, ripe for erosion and reflect storage of aeolian particles on the landscape, not a primary aeolian flux.

It would be informative to contact PI-SWERL person, in tuned with aeolian particulate sources in the SW to provide a fuller assessment of the conditions of aeolian flux, related to source, sediment availability and threshold wind speed. The far-travel dust sources should have a different mineralogic signature than local sources. There needs to be more data to support the aeolian flux premise of this paper; it is not overly convincing, at least to me.

Also, the sedimentation rates and dust flux are low to very low particularly from 30 to 12 ka, which indicates that this system has low sensitivity to aeolian flux? I question what the low variability in aeolian flux from nearly zero to ~1.5 g/cm/ka at ca. 130 ka actual reflects and is this low variability significant? It would be helpful to assess the current aeolian flux around Stoneman Lake; is this area sensitive to current dust sources? Why are the early Holocene rates (11-9 ka) of dust flux, when many pluvial lakes existed, though at declining levels, so much higher than 5e and stage 7. Something is missing in explaining these state differences in aeolian flux between Stage 1, 5 and 7, suggesting non-analogous aeolian sources and sinks.

Lastly, a recent detailed climate model of the last interglacial in the SW USA demonstrated a dichotomy in precipitation between Arizona and New Mexico/TX with Arizona much drier than to the east (Insel and Berkelhammer, 2021 in J. Quat Science), with a warmer Gulf of Mexico, potentially energizing the NA Monsoon. In this analysis there would be a dynamic boundary between moisture sources from NA Monsoon and Equatorial Pacific, particularly in the summer and how aeolian flux would be governed in the SW USA remains an open question. More data and climate modeling is surely needed.

Version 1:

Reviewer comments:

Reviewer #1

(Remarks to the Author)

Dear editor,

Herewith my review of Staley and colleagues sent to your journal for consideration.

I have seen this article before in another review round and I see now that the authors have considerably improved their work, also in response to my comments.

A main comment I had was that the regional and local dust records were discussed separately in two different papers sent to different journals. The authors have now merged these two papers considerably improving the work.

I still think the work can be very interesting to be published in your journal. Scientifically, I have no doubt about the ruggedness in which the authors have performed their data gathering, data analysis and interpretations set into the frame of scientific knowledge and data. I have worked myself in stratigraphy, paleoclimatology and including with long-term dust records, while I am not totally familiar with the Quaternary and geomorphological aspects of the study nor with the region. For a further detailed critical review on these matters, you will depend on other reviewers, which, as it seems from the previous round, is secured already.

My current opinion is that I do not think the paper writing has matured yet. My advice is to ask the authors to considerably improve the clarity of writing by focussing on the key message of each section and paragraph in simple words and by that making the article much shorter with less figures. All sections and the figures of the paper are not highlighting the key findings and key message and lack easy overview of important messages from explanation of these. Regularly, the paper reads like a review paper more than a Nature Communications paper. You may consider my opinion as to be mostly important from the scientific point, I still bring a bit more detail here to the writing style and presentation, as that is also an important aspect to the science communication which an article is. The title is too general, the abstract not bringing the key points in a direct manner, the introduction is not specific enough, the discussion is often too much written like a review and the figures bring a lot of points and data, some of which are hardly used and others could be better merged. Two model figures are brought that should be merged and simplified, as it seems, and the same seems valid for some of the data figures. This may become a key paper on the topic if the key message is brought readily and straightforwardly from the beginning (title and abstract).

Below my more detailed suggestions to the paper:

Line 1-2 It seems the "Hillslopes to dust" part of the title is a bit confusing as hillslopes can be dusty themselves, and the total title does not hold the key finding of the paper, which would make it attractive, now the title is a more a sort of review without big excitement.

Line 11 rivers are part of alluvial systems and so fluvial systems are part of alluvial systems while not all alluvial systems are fluvial. Find better wording or explain?

Line 10-32 It is needed to read the Nature Bold Paragraph guidelines to improve the abstract. There is no key research question explained rather than 'Understand the geomorphic impacts of climate change' which is too general, what exactly do we still need to know? The methodology (end-member, STL-14, sediment core) nor abbreviations (SWNA, STL) make it less attractive. Writing should be more direct and concise, rather than first explain the interpretation (line 18-19) first come with the finding (line 19-20) and then find out time is up to explain it (delete line 18-19 from the abstract) and explain the main text.

Line 25-27 this is the key finding, right? Why is it only there, not in the title, not early in the abstract? To make it attractive and clear to the reader what has been found new compared to earlier studies, bring this directly and then explain this key finding in the remainder of the paper.

Line 28-30 is there immediate and direct evidence for this in this work? Then it is also a second (subordinate?) key finding, right?

Line 30-32 could be very important, third, subordinate, key finding? Not being exactly expert in the field of the authors, while having worked quite a bit with long-term dust records, still, this is rather specialized knowledge, that could be in the abstract, but importantly then it needs more explanation or different, simpler wording. It is a general journal this is sent to, not a specialistic one.

Line 61 One could also start the introduction with line 61 and leave the first two paragraphs for a review paper, it gives space in the article for more direct text, such as, in particular also for specific missing knowledge to the general statements given in Lines 67-71. What exactly is missing due to these long-term records are 'largely' absent? What means largely absent? There are no references given with the statement, why not? There are records it suggest, what do these show?

Lines 72 - 123 these are a mix between rationale, introduction and methods. I would not write it the way it is done now, as it combines the study sites and methods with explanation of what these sites mean in terms of environment and how these records may be interpreted. But because the records are not yet there, the reader has to remember a lot of things, while the exact same reasoning will come back in the article when the data are interpreted. So, most text is duplicate. I would skip this reasoning here, refer to the methods for the study site specifics, and use the space here to provide a longer introduction to the work setting the research questions in the frame of the state-of-the-art knowledge with only a minor 'approach' paragraph.

Lines 124 Results: Most of the results are followable and logically written in order. An outsider has, however, to push all gas and release the breaks to keep up with the matter. This could still be improved by starting paragraphs with more easy phrasing of the main finding presented in that paragraph without abbreviations. As example, lines 138-140, have an explanation to something the outsider does not yet comprehend and they are thinking 'why do I read this?'. Next lines, bring the key finding of this paragraph: "46% of the total clastic sediment volume is interpreted as being eolian dust". If the latter would be the first line, the reader would know why to read the explanation of this finding in the line after that one. Another way, in general, to write more clear is to shorten paragraphs to one message per paragraph (the first line) and two-three (max. four?) lines explanation and then start a new paragraph with a new, next message, etc.

Lines 197 onwards Discussion: the Discussion suffers from the same needed improvement in writing clarity as the Results. As example, line 224-225 bring the key results of a long section of reasoning. I would turn this completely around to let the reader know directly what comes out and then explain this, particularly if this is done in the 'short-paragraph' style explained above, it will considerably help readability. It could be in steps to go through a certain reasoning, but I would not wait for the very end to come with the finding. And, turning it around likely also results in less 'review-style' writing and more to-the-point reasoning resulting in less text.

Line 228-230 is typically Introduction content and here I would come immediately with the finding: "we find higher dust accumulation during peak interglacial conditions". "This is in contrast with...". Or, "the higher dust accumulation during peak interglacial conditions in ... is in strong contrast to..." to make it ever more short and to-the-point.

Line 271 explain why it does not capture this

Figure 1 why is the well site not indicated in 1C?

Figure 3 I would think the global records can all go to Figure 5? And, instead better here to combine with data in Figure 7? All data new in the paper in one figure?

Figure 6 and 8 to be combined? These schematics are very important to such papers, but the complexity is a bit too high it seems? And the need for two is not clear to me, apart from that a lot of text is added into them, which could be done more schematically with T up (arrow) and P down (arrow). Clouds could be schematized into one cloud darker with thick rain or lighter with thin rain, to save space

Reviewer #2

(Remarks to the Author)

This manuscript offers an interesting dataset and insights that could make a significant contribution. The authors have significantly revised the previous version by adding grain-size data and details of end member analysis and have expanded the discussion to include how climate change affected source terranes and dust production. The exploration of various dust sources and their changes in response to glacial and interglacial climates is intriguing. However, while the work presents some compelling ideas, it requires reorganization to clarify the reasoning process and ideally the inclusion of new data to support key interpretations in the discussion. As currently interpreted, the dataset may not justify publication in Nature Communications. After reorganization and strengthening the reasoning with evidence, the work could make a substantial contribution to a discipline-specific journal.

Here are some specific comments:

1. The introduction and background information are currently combined in a way that causes confusion. The interpretation of sediment sources for each dust grain-size fraction is crucial to this study, yet the manuscript treats this information as background without citing any references or repeats data interpretation. This leaves readers wondering about the basis of this knowledge. I suggest adding a section titled 'Geologic Background,' where the authors can introduce the study site, describe the wind patterns, detail the bedrock compositions, potential sources, and nearby drainage systems.
2. Results seem like Results and Interpretation; it should be better streamlined.
3. The discussion that dust abundance only reflects sediment supply in source terrane is not well supported.
4. The discussion about aridity and precipitation vs. moisture in glacial vs. interglacial period is confusing.

Line 44-45, autogenic vs. allogenic processes, not clear what is specifically referred to in this paper. It will be helpful for readers if the terms in the subsequent sentences are linked directly to these two terms.

Line 64-67, this sentence is correct but does not fit in here well for the writing does not extend to explain why studying past dust record is relevant to changes induced by human activities.

Line 73, rapid climate change only refers to glacial/interglacial cycles here, be specific.

Line 133, the 5 end members are not described.

Line 138, It is confusing to lump quartz with a clay mineral and a heavy mineral zircon together. Rephrase to something as quartz accounts XX% (what is the link between quartz and basalt?), illite is the dominant clay mineral and it can not produced from weathering basalt, and abundant zircon grains can not be from basalt.

Line 140, the connection between the two sentences is not clear. How do quartz, zircon and illite link to EM3 and Em5, are they mostly in this two end members? No evidence is provided.

Line 142-143, how about flow velocity? Strong vs. weaker wind.

Line 181-190, why EM2 can not be dust with sources between those of EM3 and EM4? What is the relationship between EM5 and Ti/(zr+rb)? EM2 peak is high, what about the element ratio is mainly driven by EM2? Also can EM2 be the weathering product of EM3 and EM4 (thus ultimately dust) given that catchment soils have high EM3-5? Can EM5 be from episodic sandstorms?

Figure 3 has lots of information not mentioned in the text.

Line 207, Tau et al say that "Particle-size distributions reveal fine dust transported during winter from the northwestern Sonoran Desert and the Mojave Desert and coarse dust transported during summer from the southwestern Sonoran Desert, similar to current climate systems and dust pathways." Different from the argument that is cited for.

Line 220, Even the westerlies are stronger during glacier period, how about monsoon? McGee et al. say "We demonstrate that a wide range of data supports wind gustiness as a primary driver of global dust levels", which is exactly opposite to what the paper is cited to support.

Line 221, do we know roughness should be higher or lower during glacial period?

Line 233, appear to be influenced by global climate changes, is it specifically glacier-interglacial cycles? The reasoning flow is confusing here.

Line 245, "wetter", you said monsoon and tropical cyclones were suppressed, how come it became wetter or have increased effective moisture (what is that)?

Line 295, "Verde Valley dust" is there any geochemical evidence to support the interpretation? This is an important component of the dust generation mechanism.

Version 2:

Reviewer comments:

Reviewer #2

(Remarks to the Author)

This manuscript has undergone a significant transformation. My perception of the work and its significance has completely changed after reading this version.

The work presents a long and high-resolution dust record from southwestern North America, providing valuable insights into the complex interactions between dust generation, climate, vegetation, and erosion processes. This research is important not only for the last 200,000 years but also has implications for the study of older loess, given that loess records extend further back into geological history. The manuscript is well-written, exhibiting a logical flow of reasoning. The figures are well-crafted and enhance the reader's understanding. The methods section is sufficiently detailed, allowing readers to evaluate the uncertainty associated with the end-member analysis. Furthermore, the conclusions and interpretations are well-supported by the data.

I am very impressed with the scientific narrative and the authors' ability to convey the story. I thoroughly enjoyed reading this work and do not have any major comments, just a few minor suggestions to enhance clarity. While I hold a very positive view of this study, I would like to note that I am not a Quaternary geologist and may lack the expertise to fully evaluate the specifics of Quaternary regional climate.

1. In Figure 4, MIS 3 is indicated in light blue, please add a note in the figure caption to explain this distinction. Additionally, the three proxies from the Chinese Loess Plateau, Japan, and the Northwest Pacific appear to be primarily influenced by dust production in Central Asia, raising questions about why this is considered a global flux. The dust curve from the East Atlantic shows significant differences compared to the other three proxies. It may be worthwhile to adjust the text in the related area, replacing "global" with "other areas around the world," as the main point is that Southwest North America differs from these other regions concerning the causal relationship between climate and dustiness.

2. In Figure 6, there are several climate proxies included. Given that the text may not have enough space to explain how each proxy indicates different climate conditions to support the interpretation, it would be helpful to annotate the figure to indicate warmer/cooler and drier/wetter conditions, particularly for $\delta^{18}O$. Additionally, since the text discusses 20 kyr cycles, marking these cycles in the figure would enhance clarity.

Reviewer #4

(Remarks to the Author)

The Stoneman Lake core has an impressive ~230 kyr record of dust that has allowed Staley et al. to consider controls on glacial-interglacial timescales of dust flux. The conclusion that interglacials are dustier than glacials is an interesting one, and they link the dynamics of the eolian system back to climatic-driven changes in sediment supply. Tau et al. (2021) made a similar conclusion for Holocene dust at nearby Montezuma Well, and so it is nice to see some broad agreement, although the two papers differ on atmospheric controls and where the dust may have come from. I also think that geomorphologists

would expect that in some places, regional controls may counter global trends in dust emissions, and there is a strong case for that here. Their end member modelling identified two distinct dust modes (EM3 and EM4). They found that the EM3 and EM4 modes are not always synced, suggesting a heterogeneous landscape response to climate change and dust emissions. I like this because regional dust sources are not controlled by an on-off switch, as is assumed in some dust emission models. The long geological record of dust as well as results that counter global trends in dust are significant contributions and worthy of publication in Nature Communications.

I think the paper reads well – it gets to the point and provides the key data to support their conclusion. This is my first review and having seen the author responses to the previous reviewer comments, I sense there have been a lot of improvements in both the writing and figures. I can follow the figures that support their interpretations.

I think the models and reasoning they present to explain the different grain size populations as identified through end member modeling make sense, although I think some clarification is needed for EM4 coarse dust sources. The authors suggest that the primary source of EM4 is the Verde Valley. They have identified the location of deposits on Fig. 1. They include several citations of maps and reports documenting these deposits (citations 54-60). However, they make an assumption that the Verde River deposits were a source of dust. Reviewer 2 posed a similar concern, and the authors replied that no geochemical provenance analysis has been done to prove the linkage. Their assumption, and their model, could falter if the river was gravel dominated in this location because they present no alternative source. Significant fine grained deposits are required to warrant wind erosion and dust production. Do citations 54-60 identify fine grained fluvial deposits? If so, this is not stated directly. The authors seem to assume the river would “periodically exposing fresh fine sediments to the atmosphere” without providing evidence of the fine sediment. If 54-60 document such fine deposits, say so directly. If not, or in addition to, consider checking the soil survey for this area. I was able to see the area highlighted in yellow on Fig 1 in soilweb-apps, where some of the area is dominated by sandy loams and other fine soils at the surface. Confirming the existence of fine grained (sandy) deposits makes their model seem plausible.

Author's Rebuttal - NCOMMS-24-02264-T

Below we respond to the comments made by each referee and the Associate Editor. Comments are italicized and our responses are in regular font.

Reviewer #1's comments (italicized) and our responses:

The study has not been uniquely sent to your journal. The dataset seems to be completely sent to Geology also, where another part of the study is being discussed. This makes the current work less attractive, as the work is split into two, but also less holistic. Why would one discuss the local dust transport separate from the regional dust transport? In the end, it is the same region and one cannot be seen separate from the other? The authors wish to publish in two of the better journals in their fields, why not combine that into one great study rather than two intermediate studies?

And

Lines 172-173: meaning the same record is being published elsewhere? One comprehensive study would have been nicer as SRT and MRT are not that different?

We agree. We have incorporated the two studies; the other manuscript is no longer being considered by *Geology*. Our revised manuscript now discusses regional heterogeneity in dust emission caused by diverging responses between alluvial and fluvial dust sources to climate change. We demonstrate this by comparing our regional desert dust and local fluvial dust accumulation records side by side.

There is not a proper uncertainty propagation when the differences are calculated (standard errors) of dust fluxes between climatic stages. The 1.6 times change in flux should be going along with a standard error to know what is its significance. This nor any of the other 'differences' nor standard errors are quantified in the submitted paper. The key figure providing fluxes per stage is rather small and not sufficiently explanatory (see below minor comments).

And

Line 16: it is needed to add the standard error of the difference to this number, such to see the relevance of this quantification within the total uncertainty of the data.

Line 172-173: We have calculated a range for the change in total dust flux between glacial maximum (MIS 6 and 2) and interglacial maximum (MIS 5e and Holocene). The range is based on the calculated error envelope: minimum of range = [median of minimum values during interglacial maximum]/[median of the maximum values during glacial maximum]. Maximum of range = [median of maximum values during interglacial maximum]/[median of the minimum values during glacial maximum]. Figure 4 is improved and presents the notched box and whisker of median values in each timeseries. Notches that do not overlap indicate medians differ with 95% confidence.

The Results section jumps into interpretations from the second line, while many results are not described and only come in the Discussion section. The Discussion section has a too high degree of certainty given in words to interpretations. As most interpretations come back in the Discussion anyway, it is needed to clean the Results section from interpretations and appropriately describe interpretations as possibilities in the Discussion section.

And

Line 73: The second sentence of the Results section brings an interpretation. It would be better to first describe the results.

And

Line 71: please start the results section with a basic description of the main results, namely Laser Granulometric results of the samples.

And

Lines 122-125: here it is needed to present all results: Figure 4 is only partially presented and the key number from the abstract including its uncertainty does not figure in the Results.

And

Line 127: The phrasing is not scientific. EM-3 is an end-member peak in a end-member modelling scenario representing the 13 micron grain size mode. EM-3 could relate to Medium-range transport dust, that, if that would be the case, is likely of North-American origin. It is an interpretation, not a fact. The authors have to rewrite the text making clear distinction between facts (a measurement, 13 micron mode) and an interpretation (MRT, North American source).

Our results are now more thoroughly described. We also clarify our reasoning for interpreting sources for each end member and are more careful about distinguishing facts and interpretations. We have also included more context in the form of previous mineralogical and geochemical results (Lines 148-154) that support our end-member interpretations. Within this context, we believe these initial interpretations of source are robust enough to be included in the results and allow our discussion to be more process oriented.

The Discussion section dwells upon details of functioning how it all may have happened, the model, while I miss a much more straightforward line of reasoning from data to simple interpretations. Some claims further in the Discussion section are more 'review paper' like than that these relate directly to data in the current paper, also because the 'other half' of the data are not discussed here. The authors may have gone much further in their reasoning and describe that, while forgetting that the reader has to do it with what is currently provided. A more direct and concise Discussion chapter will help the paper.

And

Lines 260-263: that seems a very straightforward conclusion that one may derive from the data, why was it not brought forward earlier? In the relatively complex discussion before this part, there was very little discussion about vegetation.

And

Lines 269-274: higher effective moisture means more vegetation cover and less exposure and so lower dust transport. I do not see why the current data are adding to the specific discussion of climatic perturbations other than that reasoning. It seems the discussion is rather complex for a not such a difficult case?

The discussion has been simplified and streamlined.

For a wider audience, it is needed to write with much less abbreviations and with less assumptions as if the reader would know things, see minor comments. The reader for Nature Communications should be regarded as an interested geoscientist, nothing less and nothing more. Different writing is needed for that.

And

Line 111: The authors should use less abbreviations in the running text: DMAR, MRT, SRT, etc. and in particular not without explanation in each paragraph. The journal is aimed at a wider audience than specialists. This whole paragraphs reads great for those involved in the study, but is

impossible to read for a (relative) outsider. This could also include finding other words for these items, MRT does not even include the ‘dust’.

And

Lines 159-160: If MRT is renamed to desert dust, why not initially starting with that name, in my view, better than MRT as more explanatory, and better than using two names in the running text.

We appreciate the comment about broad reader accessibility. We believe we have addressed this and have replaced most abbreviations with full text.

Lines 81-85: EM-2 would be perfectly in line with long-range dust source, the reasoning here seems a bit weak?

And

Line 104-105 Ruling far distance dust transport is a bit dangerous, I would think. First, what about relating EM-2 to long distance transport?

Yes, the EM-2 mode (5 μm) is consistent with LRT, but its accumulation trends and absolute rates are not. Asian dust is the most likely source of LRT for North America. If we compare flux trends of Asian dust to those of EM-2 in Fig 3d and Methods Fig. S3c, EM-2 is only high during early interglacials and ranges from $\sim 1\text{-}10 \text{ g/cm}^2/\text{kyr}$. Conversely Asian dust is lower during interglacials and flux to the NW Pacific (still 1000's of km upwind of Stoneman Lake) never gets higher than $1 \text{ g/cm}^2/\text{kyr}$. Long-range transport of dust from Asia and Africa certainly occurs, we see it in the modern, however, it occurs in quantities that would be barely perceptible here compared to influx of medium- and short-range dust and local alluvium. We explain this reasoning in lines 181-190.

Line 132: write in full STL

Stoneman Lake is now written out in full where it appears.

Lines 143-145: please explain “cannot be controlled by intervening processes of transport and deposition”. Who controls what? What is intervening what? Is the MRT interpretation impossible to validate?

Clarified in lines 210-226.

Line 155: it would help if the authors write more to a general audience also by more explicit explanations. Here it is assumed that people know what is ‘wet removal’, that the glacials have the highest effective moisture.

Clarified in lines 212-213. We cite lake depth in Fig. 3f to support that glacials have highest effective moisture (lines 221-224).

Lines 154-156: isn’t this partially a circular reasoning? First, it is assumed that a certain grain size class is purely dust, and then, because there is higher dust flux, there is not much wet removal?

Lines 138-159: We believe our interpretation that EM 3 and 4 are dust is strong enough that the reasoning here isn't circular. Geomorphology and geochemistry/mineralogy of soils in the lake catchment has been done in a previous study (Staley et al., 2023). We showed that both soils and lake sediments are a mixture of locally sourced minerals (dominated by albite, smectite and

kaolinite, and ilmenite) and dust (quartz, illite, and zircons). We have tracked local inputs using Ti (from ilmenite), fine dust using Rb (from illite), and coarse dust using Zr (from zircons). The geochemical ratio tracing the relative contribution of local derived material to dust (Ti/Zr+Rb) parallels that of EM 1&2&5 to EM 3&4 (Fig. S5). Likewise, the geochemical ratio tracing the relative contribution of coarse to fine dust (Zr/Rb) parallels that of EM 4 to EM 3. Our interpretation is further supported by EM 3 (13 μm) and EM 4 (38 μm) being nearly exact matches of end members (13 and 34 μm) in a nearby study (Tau et al., 2021) interpreted as fine and coarse dust.

Line 159 SWNA?

SWNA is defined in the introduction. We can write it out in full, or as SW North America (like the title) word count permitting.

Line 159: Evidence ‘dictates’, not sure if that is really the appropriate academic wording.

Corrected to suggests. Line 224

Line 162: I propose to include the grainsize mode also in between brackets, SRT, EM-4, 38 micron). Particularly the grainsize is what helps the reader to memomize the data and grains, EM-4 is different for every study, as is SRT.

We have incorporated this suggestion where appropriate.

Lines 163-164: this should be in the Results. And, looking at Figure 4, there is no mention of SRT, so where should I look?

Moved to results (line 160-177). EM 4 (SRT/Verde Valley dust) is a significant part of this revised manuscript and discussed in lines 284-341.

Lines 166-167 Please explain “whose published depositional ages correlate to MARsrt increases”.

Clarified. Lines 298-300 and Fig. 7b

Line 177: SWNA in the title is not very explanatory to a wider audience

Changed to SW North America (line 221)

Lines 208-209: I do not see this in Figure 3: the transition from MIS 6 to 5 is rapid in the Desert Dust curve given, where is the 20kyr lag?

Clarified. Lines 258-261 and Fig. 5a

Lines 232-233: Provide the reader means to find this in the figure, you mean to look at c, d, and e? Relating the tiny spike at 7>6 transition seems too far, the overall agreement in trends between b and c/d/e is, however, worthwhile more reference.

Part about brief spike removed. Substage a, b, c, d trends are discussed in lines 262-274 and their boundaries are now indicated in Fig. 5.

Line 258: The lack of a sustained vegetation cover in glacial times is a primary reason for increased dust fluxes, or do the authors include that in the ‘dust production’?

Thank you for the comment – in fact this is the opposite and we need to clarify this in the discussion. Glacials = more veg/less runoff = less dust (Lines 228-247).

Lines 260-263: there must be vegetation records for comparison to corroborate this claim?

There are. We have included a reference to Betancourt (1990) in Line 246.

Lines 266-267: Figure 5 gives a clear lower STL MARsrt for the mid-Holocene compared to Early Holocene and Late Holocene, that is not sufficient?

See Lines 275-283 and Fig. 5a

Figure 2: why are the real data hidden behind the end-member analyses? Choices for line thickness are such that the real data are not easy to trace in the figure.

We have addressed this in Fig. 2 with an a panel showing raw data and b panel showing end members.

Figure 4: I find this figure / table hard to read, it could be my age, but it all seems rather small, particularly because the red outliers need so much space that the averages and 2.7sigma intervals are somewhere near the bottom of each panel. Would it be an option to leave the outliers for a figure in the appendix and blow up this panel focussing on the main box and whisker plots?

We have incorporated your suggestions and focused Fig. 4. Thank you.

Figure 4: I cannot find the paper nor Methods section what intervals are included in ‘Glacial’ and what in ‘Interglacial’. Same for Early, Middle, Late, add a description in the text / caption. It would help to indicate early, middle, late also to Figure 3 on the time axis. While dividing into glacial / interglacial seems evident, it may not necessarily be. One could argue to only compare Stage 1 and 5e with Stages 2,3, and 6. Or more independent, a ‘cutoff’ could be used on the LR04 isotope curve to decide which data point bins in what stage, glacial or interglacial. It may make the results more significant and definitely more independent.

Clarified in the Figure 4 caption. Figure 4 has also been streamlined to highlight the most important intervals. Glacials and interglacials are defined in Lines 164-169. Early, middle, and late Holocene boundaries are indicated in Figs. 3, 7, and 9.

Figure 4: labels ‘e’, ‘abcd’ are unclear, these all belong to MIS5? It seems there is space on the image to show that.

Figure 4 has been streamlined so data on MIS 5abcd is no longer presented.

Reviewer #2's comments (italicized) and our responses:

Introduction: I do not see the tight connection of the research topic and justification of scientific importance in introduction. Ice cores and deep sea are great proxies for global dust budget. The importance of American southwest to the global picture is quite vague here, and there is no critical knowledge gap or debates described to clarify the contribution of this study. The human impact on dust generation also seems not relevant given the study extends back to the late Pleistocene. I see the importance at the end of the discussion, but here it is not appropriately sold.

Thank you for this perspective. We have revamped the introduction considering we now have included the full dataset in this revised manuscript (we have incorporated the manuscript that was sent to *Geology*. Now only this manuscript is under consideration). In a nutshell, SWNA lacks continuous and long-term dust and geomorphic activity records that may shed light on potential outcomes of rapid climate change. To the best of our knowledge, this study is the first of its kind in North America to continuously quantify both dust accumulation and hillslope sediment transport rates over glacial-interglacial timescales.

Background: the text lacks key description of the background (description of figure 1), particularly the climate and atmospheric circulation patterns.

We have added this. See Lines 72-109.

Results: 1) why choose five end members? Is this based on observation and sound geologic interpretation or just math technique? I think it is the latter, which, in my opinion, does not have geologic significance. I understand there is knowledge of modern dust source (like Tau et al., geology) that this study is based on, but the reasoning is not well clarified. 2) The explanation of each end member seems very casual. For example, how physical and chemical weathering affect the minerals ton have EM1 and EM5? I know the reference is cited, but such interpretation is the core of the study and should be clarified here. 3) if EM 1 coalesces with EM2, what is the basis to separate them?

And

How does the geochemical data support? It needs some clarification.

We have clarified our model choice in Results (Lines 125-137) and in Methods (Lines 629-669). This choice is based both on how the mathematical model fits the observed data and geologic interpretation. We clarify the geochemical/mineralogical support for our chosen model in Lines 138-159; 178-190. Below is a summary of our reasons to choose the 5 end-member model scenario and which sediment populations they represent.

AnalySize, our end member modeling software of choice, automatically runs multiple scenarios that fit 1 to 10 end members to the measured grain size dataset. It returns model performance statistics along with PSD and percent composition in each sample for each modeled end member (Fig. S2). The best end member model scenarios for further examination were those that prescribed 3, 4, and 5 end members (3EM, 4EM, and 5EM) because they have a total model fit r^2 of ~96, 98, and 99%, respectively, and exhibit little intercorrelation between constituent end members that would indicate model overfitting (Fig. S2ab). The first end member of the 3, 4, and 5 end member scenarios (3EM-1, 4EM-1, and 5EM-1) has a primary mode at 0.18 μm and one to two other secondary modes in the clay to silt range (Fig. S2cde). 3EM-1, 4EM-1, and 5EM-1 also exhibit nearly

identical percent composition by depth (Fig. S2cde). The poly-modality of 3EM-1, 4EM-1, and 5EM-1 is consistent with locally sourced sediments (they are poorly sorted) and consistent results between scenarios indicates robust modeling. Moving on, 3EM-3, 4EM-4, and 5EM-4 are also similar in grain size distribution (single modes of 31, 42, and 38 μm) and stratigraphic variation (Fig. S2cde). These modes are consistent with short-range-transport dust (dust from <100 km away) and a coarse dust end member (mode: 34 μm) identified in sediments of Montezuma Well, AZ (25 km to the SSW of STL), a lake that is downwind of the same dust sources (Tau et al., 2021). The main difference between the 3EM, 4EM, and 5EM scenarios is how they model sediment populations in the very fine to fine silt range. Unimodal endmembers in the fine silt range are consistent with medium-range-transport dust (dust from 100's of km away). 3EM-2 has a unimode centered at 9.5 μm (Fig. S2c). If we interpret 3EM-2 and 3EM-3 as dust then there is no input of local alluvium during interglacials MIS 7 or MIS 5e (Fig. S2c) which is contrary to geochemical and palynological evidence indicating quite vigorous sediment transport during these times (Jiménez-Moreno et al., 2023; Staley et al., 2023). In the 4EM scenario, particles in the very fine to fine silt range are split into finer (4EM-2: 6 μm) and coarser (4EM-3: 13 μm) end members (Fig. S2d). 4EM-2 is interpreted as resuspended fine sediment from the lake margin (e.g., Arcusa et al., 2020), likely very abundant after lake levels fall during early interglacials, explaining high abundances rates following glacial to interglacial transitions during MIS 7, 5e, and the Holocene (Fig. S3). 4EM-3 is an exact match of the fine dust end member (13 μm) from Tau et al. (2021). We ultimately chose the 5EM scenario because it is nearly identical to the 4EM scenario while adding a fifth (primary mode at 169 μm ; secondary mode at 19 μm) that improves model fit during a few time periods around 31 and 12 ka (Fig. S2e) when coarser material intercalates the lake stratigraphy. 5EM-5 is another locally sourced sediment population given its coarse and poly-modal PSD. In sum, the 5EM scenario maximizes the variance accounted for in the model (Fig. S2ab and S4) and produces end members whose particle size distributions compare favorably to those of other regional dust studies and whose accumulation trends stand up to geologic interpretation.

Further support for the 5EM model and interpretations of end members is provided by comparison to mineralogical and geochemical analyses by Staley et al. (2023) of soils in the Stoneman Lake catchment the very same lake sediment stratigraphy studied here (Fig. S5). We showed that both soils and lake sediments are roughly a 50-50 mixture of locally sourced minerals (dominated by albite, smectite and kaolinite, and ilmenite) and dust (quartz, illite, and zircons). We tracked local inputs using Ti (from ilmenite), fine dust using Rb (from illite), and coarse dust using Zr (from zircons). The geochemical ratio tracing the relative contribution of locally derived material to dust (Ti/Zr+Rb) parallels that of EM 1&2&5/EM 3&4 (Fig. S5). Likewise, the geochemical ratio tracing the relative contribution of coarse to fine dust (Zr/Rb) parallels that of EM 4/EM 3.

Discussion: Section 1: 1) the evidence to rule out “processes of transport and deposition” is not clear; 2) Westerlies may be stronger during glacier maximum, what about monsoon? The evidence ruling out wind intensity change is also not compelling. 3) cite reference and evidence for that the highest effective moisture was during glaciers.

- 1) We do not see increases in dust accumulation during periods we would expect to see transport (wind speed) and deposition (surface roughness and precipitation scavenging) processes positively affect accumulation rates (glacials). See Lines 210-226 for clarification.
- 2) The monsoon question is interesting because strong convective cells likely play a role in dislodging and exposing fine sediment to wind erosion. However, according to Metcalfe et al.

(2015) the NAM appears to strengthen during the mid-Holocene and this is not associated with an increase in dust accumulation in our record.

3) Line 219- Highest effective moisture indicated by lake depth (Fig. 3f).

4) evidence for the different source of SRT and MRT?

We interpret different sources for SRT (EM 4, now called Verde Valley dust) and MRT (EM 3, called desert dust) based on their different grain size modes and accumulation trends. See earlier response.

5) Anderson et al say that “fluvial aggradation beginning during full-glacial times and continuing into subsequent interglacial”, is this consistent with the observation of SRT?

Yes, see Lines 322-331.

Section 2: line 177-179, this observation is limited to mid-latitude, right? How about desert, why that is not included given desert to the north and northwest of China loess plateau has been suggested to be a source.

This observation is for SW North American deserts and we have ruled out Asian sources. Clarified Lines 252-254.

Line 181, the transition from alluvial source to desert is very confusing.

Alluvial sources are found in the desert. This is clarified in Lines 248-261.

Line 182, if I understand right, this is for MRT, why not replace “fine” with the grain size for clarity.

This part of the manuscript was removed, and efforts were made to clarify the different modes of each EM where appropriate.

Any evidence of production of this grain size range at desert hillslopes?

Frost weathering, fluvial and alluvial comminution, and erosion of siltstones are all capable of producing silt sized grains from desert hillslopes (Smith et al., 2002; Muhs, 2013). MRT (“desert dust”/EM-3: 13 um) is consistent with regional dust in both modern and paleo records (Reynolds et al., 2016; Routson et al., 2016; Arcusa et al., 2020; Tau et al., 2021).

line 194 and 197, it is not clear what are the climate-driven surface processes, and how they were controlled by climate and ecological changes.

This discussion has been completely rewritten. See Lines 228-283.

Line 209, I thought with the stage set up earlier in this paragraph, here you want to talk about the slope instability during the interglacial, not sure why you change to talk fluvial adjustment to sediment load increase?

This discussion has been completely rewritten. See Lines 228-283.

Line 261-263, this seems different from what has been said earlier. Is this a comparison of glacier maxima to the rest of the glacier periods? I think the information provided so far is that glacier is less dustier than interglacial.

Clarified in Lines 237-247. We discuss glacials as a whole, not variability within glacials.

I think the rest of the discussion is good and interesting, but they are based on discussion in the first two sections. The lack of evidence and clarity in the first two sections undermines the extended discussion.

We hope to have added enough clarity and justification of our interpretation of end members to allow this discussion, which has itself undergone significant revision.

Reviewer #3's comments (italicized) and our responses:

Key points:

The central inconsistency of this paper is the lack of granulometry data from the surrounding small catchments to identify local water-transported flux of particles versus aeolian dust flux. There needs to be a detailed study of the source to sink flux of particles from numerous sources around the lake including catchments, overland flow, and aeolian proximal and distal sources. What is the geomorphology and soils around the lake? Vesicular A horizons, a common desert soil morphology, are loaded with aeolian particles, ripe for erosion and reflect storage of aeolian particles on the landscape, not a primary aeolian flux.

Geomorphology and geochemistry/mineralogy of soils in the Stoneman Lake catchment was accomplished in a previous study (Staley et al., 2023). Soil storage is limited to glacial maxima (Fig. 7d) as the catchment is small, steep, and dominated by alluvial/colluvial soils categorized as argiustolls (no Av horizons). Furthermore, the lake surface itself accounts for ~20-25% of the basin's total area suggesting that even during glacial maxima, a primary dust signal is still present.

Lines 138-159. Both soils and lake sediments are an even mixture of locally sourced minerals (dominated by albite, smectite and kaolinite, and ilmenite) and dust (quartz, illite, and zircons). We have tracked local inputs using Ti (from ilmenite), fine dust using Rb (from illite), and coarse dust using Zr (from zircons). The geochemical ratio tracing the relative contribution of locally derived material to dust (Ti/Zr+Rb) parallels that of EM 1&2&5 / EM 3&4 (Fig. S5). Likewise, the geochemical ratio tracing the relative contribution of coarse to fine dust (Zr/Rb) parallels that of EM 4 / EM 3.

We have included granulometry of soil samples from a fairly flat and small surface just north of the catchment (Fig. 2a). It has a similar particle size distribution as lake sediment and therefore seems to be a mix of similar end members (local weathering products and dust).

It would be informative to contact PI-SWERL person, in tuned with aeolian particulate sources in the SW to provide a fuller assessment of the conditions of aeolian flux, related to source, sediment availability and threshold wind speed.

Numerous studies indicate that active alluvial fans and river floodplains are the primary sources of eolian sediment in the southwest (Sweeney et al., 2013; Bullard and Livingstone, 2002 and references therein). Threshold windspeeds are very likely met over the longer (decadal-centennial to millennial) timescales that concern this study (Reheis, 2006).

The far-travel dust sources should have a different mineralogic signature than local sources.

There is a distinct mineralogical difference. Dust is dominated by quartz and illite. Local sediments are dominated by albite and ilmenite. See Staley et al. (2023) and Lines 138-159.

There needs to be more data to support the aeolian flux premise of this paper; it is not overly convincing, at least to me.

See lines 138-196

- 1) Mineralogy of lake sediments indicates the clastic fraction of lake sediments is ~50% quartz, illite, and zircon and ~50% ilmenite and albite (Staley et al., 2023). Quartz, illite, and zircon must be allochthonous given the catchment's alkali basalt bedrock. End member modeling reflects this even split with dust end members (EM 3 and 4) accounting for 46% of the total clastic sediment column analyzed and locally sourced end members accounting for 54%. Lines 138-154; 178-190.
- 2) We have tracked local inputs using Ti (from ilmenite), fine dust using Rb (from illite), and coarse dust using Zr (from zircons). The geochemical ratio tracing the relative contribution of locally derived material to dust (Ti/Zr+Rb) parallels that of EM 1&2&5 / EM 3&4 (Fig. S5). Likewise, the geochemical ratio tracing the relative contribution of coarse to fine dust (Zr/Rb) parallels that of EM 4 / EM 3. Lines 152-154; 187-190.
- 3) EM 3's 13-um mode and unimodal particle size distribution is consistent with dust that has been transported 100's of km. A similar study done 25 km SSW has a 13-um mode end member interpreted as regional dust (Tau et al., 2021). Presumably these two basins are downwind of the same sources. EM 3's accumulation rate correlates with climatic/ecologic destabilization and movement of hillslope sediment stores onto downslope alluvial systems across the SW North American deserts. Active alluvial systems are recognized as a major source of dust.
- 4) The 38-um mode and unimodal particle size distribution of EM-4, which we interpret as the coarse dust component, is consistent with dust that has been transported < 100 km. Its accumulation rate correlates to aggradational episodes of the Verde River and its axial alluvial fans. Both recognized as major sources of dust.

Also, the sedimentation rates and dust flux are low to very low particularly from 30 to 12 ka, which indicates that this system has low sensitivity to aeolian flux?

I think this suggests that flux is low, not that the system is insensitive. The lake surface accounts for ~20-25% of the basin's total area suggesting a primary dust signal is always present.

I question what the low variability in aeolian flux from nearly zero to ~1.5 g/cm/ka at ca. 130 ka actual reflects and is this low variability significant?

There is actually high variability in eolian flux starting at this climate transition ca. 130 ka and it probably relates to activation of regional sources each with slightly different thresholds for change. We interpret the lower flux and variability prior to that to reflect stable land surfaces during the previous glacial (high vegetation coverage and low runoff) (Lines 237-261).

It would be helpful to assess the current aeolian flux around Stoneman Lake; is this area sensitive to current dust sources?

There is broad similarity between modern and paleo dust accumulation rates observed here and at other sites throughout the region (Fig 4ae).

Why are the early Holocene rates (11-9 ka) of dust flux, when many pluvial lakes existed, though at declining levels, so much higher than 5e and stage 7. Something is missing in explaining these state differences in aeolian flux between Stage 1, 5 and 7, suggesting non-analogous aeolian sources and sinks.

We do not believe that the dust fluxes are substantially different amongst these interglacials as they are the same order of magnitude. Given the larger number of age tie-points in the Holocene relative to MIS 5 or MIS 7, the Holocene will appear to have greater variability in flux rates. With a necessarily smoothed age model for MIS 5 and 7, the dust fluxes will appear steadier being averaged over longer time periods.

Lastly, a recent detailed climate model of the last interglacial in the SW USA demonstrated a dichotomy in precipitation between Arizona and New Mexico/TX with Arizona much drier than to the east (Insel and Berkelhammer, 2021 in J. Quat Science), with a warmer Gulf of Mexico, potentially energizing the NA Monsoon. In this analysis there would be a dynamic boundary between moisture sources from NA Monsoon and Equatorial Pacific, particularly in the summer and how aeolian flux would be governed in the SW USA remains an open question. More data and climate modeling is surely needed.

That is an interesting point. Unfortunately, our data does not resolve specific atmospheric vectors responsible for dust transport. All we can say with confidence is that dust is reaching the SW edge of the Colorado Plateau from points south and west following prevailing zonal (winter-spring) and/or meridional (summer-fall) wind directions.

Author's Response - NCOMMS-24-02264B

Comments are italicized and our responses are in regular font.

Comments from the Editor:

"...the writing needs improvement and more information about the geologic background and the source to sink flux of the particles is needed."

The manuscript has been nearly completely restructured and rewritten with a focus on clarity including important aspects of the geologic background. For instance, Figure 1 now includes a schematic of Northern Hemisphere dust transport to provide a global context for the study and a map of regional Quaternary deposits illustrating the broad distribution of potential dust sources in southwestern North America. Key features of the Stoneman Lake catchment that make it an ideal dust trap are retained at the end of the Introduction (lines 52-55) and previous mineralogic evidence for significant dust in the record is clarified early in the Results section (lines 68-74). Our reasoning for which source-to-sink processes are primarily responsible for flux rate variability, i.e., emission vs. transport vs. deposition, has been clarified in the Discussion (lines 291-316).

Comments from Reviewer #1:

My current opinion is that I do not think the paper writing has matured yet. My advice is to ask the authors to considerably improve the clarity of writing by focussing on the key message of each section and paragraph in simple words and by that making the article much shorter with less figures. All sections and the figures of the paper are not highlighting the key findings and key message and lack easy overview of important messages from explanation of these. Regularly, the paper reads like a review paper more than a Nature Communications paper. You may consider my opinion as to be mostly important from the scientific point, I still bring a bit more detail here to the writing style and presentation, as that is also an important aspect to the science communication which an article is. The title is too general, the abstract not bringing the key points in a direct manner, the introduction is not specific enough, the discussion is often too much written like a review and the figures bring a lot of points and data, some of which are hardly used and others could be better merged. Two model figures are brought that should be merged and simplified, as it seems, and the same seems valid for some of the data figures. This may become a key paper on the topic if the key message is brought readily and straightforwardly from the beginning (title and abstract).

We very much appreciate the comments and suggestions provided by Reviewer #1, especially with regard to improving the writing and narrative flow of the manuscript. Following their suggestions, the manuscript has been nearly completely restructured and rewritten with a focus on clarity. This includes revising the Title and Abstract to focus on the key finding(s) of the study and streamlining the Introduction. In addition, we have made the Results and Discussion sections more distinct. The Discussion now maintains focus on interpreting the results, only providing the context, rather than review, of previous research.

Reviewer #1's comments on the figures led to great improvements. Figures were significantly updated with multiple mergers and restructurings meant to make the figures more concise and highlight key findings. For instance, the two model figures were merged into one, as have the figures presenting the accumulation rates for fine dust, coarse dust, and local alluvium, see below.

Line 1-2 It seems the "Hillslopes to dust" part of the title is a bit confusing as hillslopes can be dusty themselves, and the total title does not hold the key finding of the paper, which would make it attractive, now the title is a more a sort of review without big excitement.

Title has been changed to refer to the key finding of the paper.

Line 11 rivers are part of alluvial systems and so fluvial systems are part of alluvial systems while not all alluvial systems are fluvial. Find better wording or explain?

This sentence was removed.

Line 10-32 It is needed to read the Nature Bold Paragraph guidelines to improve the abstract. There is no key research question explained rather than 'Understand the geomorphic impacts of climate change' which is too general, what exactly do we still need to know? The methodology (end-member, STL-14, sediment core) nor abbreviations (SWNA, STL) make it less attractive. Writing should be more direct and concise, rather than first explain the interpretation (line 18-19) first come with the finding (line 19-20) and then find out time is up to explain it (delete line 18-19 from the abstract) and explain the main text.

and

Line 25-27 this is the key finding, right? Why is it only there, not in the title, not early in the abstract? To make it attractive and clear to the reader what has been found new compared to earlier studies, bring this directly and then explain this key finding in the remainder of the paper.

and

Line 28-30 is there immediate and direct evidence for this in this work? Then it is also a second (subordinate?) key finding, right?

and

Line 30-32 could be very important, third, subordinate, key finding? Not being exactly expert in the field of the authors, while having worked quite a bit with long-term dust records, still, this is rather specialized knowledge, that could be in the abstract, but importantly then it needs more explanation or different, simpler wording. It is a general journal this is sent to, not a specialistic one.

Following these suggestions, the abstract has been rewritten more concisely and highlights the key findings of the paper.

Line 61 One could also start the introduction with line 61 and leave the first two paragraphs for a review paper, it gives space in the article for more direct text, such as, in particular also for specific missing knowledge to the general statements given in Lines 67-71. What exactly is missing due to these long-term records are 'largely' absent? What means largely absent? There are no references given with the statement, why not? There are records it suggest, what do these show?

Thank you for this comment. The introduction has been streamlined considerably with specific references to important knowledge gaps and how this research helps to fill them.

Lines 72 - 123 these are a mix between rationale, introduction and methods. I would not write it the way it is done now, as it combines the study sites and methods with explanation of what these sites mean in terms of environment and how these records may be interpreted. But because the records are not yet there, the reader has to remember a lot of things, while the exact same reasoning will

come back in the article when the data are interpreted. So, most text is duplicate. I would skip this reasoning here, refer to the methods for the study site specifics, and use the space here to provide a longer introduction to the work setting the research questions in the frame of the state-of-the-art knowledge with only a minor 'approach' paragraph.

As mentioned above, the Introduction was streamlined considerably based on these comments. Much of the information provided does indeed come back later in the manuscript after our data is presented. We eliminated this unnecessary repetition.

Lines 124 Results: Most of the results are followable and logically written in order. An outsider has, however, to push all gas and release the breaks to keep up with the matter. This could still be improved by starting paragraphs with more easy phrasing of the main finding presented in that paragraph without abbreviations. As example, lines 138-140, have an explanation to something the outsider does not yet comprehend and they are thinking 'why do I read this?'. Next lines, bring the key finding of this paragraph: "46% of the total clastic sediment volume is interpreted as being eolian dust". If the latter would be the first line, the reader would know why to read the explanation of this finding in the line after that one. Another way, in general, to write more clear is to shorten paragraphs to one message per paragraph (the first line) and two-three (max. four?) lines explanation and then start a new paragraph with a new, next message, etc.

This is a great point. We now include simple topic sentences before each paragraph in Results aimed at enhancing the clarity and flow of the section.

Lines 197 onwards Discussion: the Discussion suffers from the same needed improvement in writing clarity as the Results. As example, line 224-225 bring the key results of a long section of reasoning. I would turn this completely around to let the reader know directly what comes out and then explain this, particularly if this is done in the 'short-paragraph' style explained above, it will considerably help readability. It could be in steps to go through a certain reasoning, but I would not wait for the very end to come with the finding. And, turning it around likely also results in less 'review-style' writing and more to-the-point reasoning resulting in less text.

Similar to the Results section, we now include topic sentences at the outset of each paragraph of the Discussion. We believe the narrative flow of the Discussion has benefitted greatly.

Line 228-230 is typically Introduction content and here I would come immediately with the finding: "we find higher dust accumulation during peak interglacial conditions". "This is in contrast with...". Or, "the higher dust accumulation during peak interglacial conditions in ... is in strong contrast to..." to make it ever more short and to-the-point.

Upon revisions, this point is now within a topic sentence of a paragraph found earlier on in the Discussion (lines 144-145).

Line 271 explain why it does not capture this

This is now clarified (lines 208 -209).

Figure 1 why is the well site not indicated in 1C?

Figure 1d now includes the core recovery site.

Figure 3 I would think the global records can all go to Figure 5? And, instead better here to combine with data in Figure 7? All data new in the paper in one figure?

All global records and the total dust accumulation rate time series from Stoneman Lake are included in a much neater and more concise Figure 4. Accumulation rates of fine dust, coarse dust, and local alluvium are combined into a single figure, Figure 6.

Figure 6 and 8 to be combined? These schematics are very important to such papers, but the complexity is a bit too high it seems? And the need for two is not clear to me, apart from that a lot of text is added into them, which could be done more schematically with T up (arrow) and P down (arrow). Clouds could be schematized into one cloud darker with thick rain or lighter with thin rain, to save space

The two schematics have now been combined into Figure 7 and schematized with less text overall. We think this made the figure more accessible and impactful.

Comments from Reviewer #2:

This manuscript offers an interesting dataset and insights that could make a significant contribution. The authors have significantly revised the previous version by adding grain-size data and details of end member analysis and have expanded the discussion to include how climate change affected source terranes and dust production. The exploration of various dust sources and their changes in response to glacial and interglacial climates is intriguing. However, while the work presents some compelling ideas, it requires reorganization to clarify the reasoning process and ideally the inclusion of new data to support key interpretations in the discussion. As currently interpreted, the dataset may not justify publication in Nature Communications. After reorganization and strengthening the reasoning with evidence, the work could make a substantial contribution to a discipline-specific journal.

Thank you to Reviewer #2 for the helpful comments and suggestions, especially regarding increasing the clarity of certain sections of the manuscript. We hope that improvements to the narrative structure, reasoning, and additional focus on the global context have enhanced the impact of this study.

The introduction and background information are currently combined in a way that causes confusion. The interpretation of sediment sources for each dust grain-size fraction is crucial to this study, yet the manuscript treats this information as background without citing any references or repeats data interpretation. This leaves readers wondering about the basis of this knowledge. I suggest adding a section titled 'Geologic Background,' where the authors can introduce the study site, describe the wind patterns, detail the bedrock compositions, potential sources, and nearby drainage systems.

This comment was key in our improvement of the manuscript. Per requests from both reviewers, we have significantly streamlined the Introduction and Results sections. Much of the study site's geologic background in the previous version of this manuscript has been removed from the Introduction. In the revised manuscript, we only include the most important aspects of the catchment that make it an ideal dust trap in the final introductory paragraph. In the Results section, the local bedrock is discussed within the context of previous mineralogical data that indicates the significant presence of dust and justifies our use of parallel geochemical data to assign sources for each sediment end-member. Remaining information about the regional setting, like dust sources and transport pathways, are presented in the Discussion alongside associated data interpretations. We believe this has resulted in large improvements and limited repetition of certain pieces of information.

Results seem like Results and Interpretation; it should be better streamlined.

In our revision, most interpretations that were made in the Results section have been moved to the Discussion. Our attributions of end member populations to fine dust, coarse dust, and local alluvium remain because we believe it is well justified in the context of previous dust/erosion studies and mineralogical/geochemical relationships.

The discussion that dust abundance only reflects sediment supply in source terrane is not well supported.

This section (lines 292-318) has been restructured to better justify and clarify. It is important to note that we are not making the argument that wind strength and wet vs dry fallout processes play no role in dust accumulation downwind, just that we find little evidence that they are the primary driver of variability in the two dust end members. To better support our argument that wind strength or particular atmospheric phenomena are not controlling factors, we now propose and discuss the support for an alternative interpretation. In this scenario, fine dust (EM3) originates from the west where sources are slightly more distant and coarse dust (EM4) originates from the south where sources are slightly closer, a scenario that mirrors the interpretation of Tau et al. 2021 (see below for further discussion on our differences of interpretation with this study). The westerly and southerly dust pathways would then be controlled by strengths of their associated synoptic atmospheric circulation, westerlies and monsoons, respectively. Westerly circulation displaces the monsoon during glacials (Bhattacharya et al., 2017), so we would expect westerly dust (fine: EM3) to increase relative to southerly dust (coarse EM4). We do not see support for this interpretation in our dataset as the opposite pattern is observed.

The discussion about aridity and precipitation vs. moisture in glacial vs. interglacial period is confusing.

We have made appropriate clarifications throughout the manuscript. Glacial periods are wetter than interglacial periods in southwestern North America, it's what makes this region distinct when it comes to glacial-interglacial dustiness trends! Perhaps there was some confusion about our discussion of strengthened monsoon during interglacials. The North American Monsoon does not

drive first order glacial-interglacial moisture variability; it is not strong enough to create long-lived pluvial conditions like those experienced during glacials. Instead, we argue its main effect with respect to dust appears to be its association with convective storms that produce intense runoff events that can erode/re-expose sediments and cold outflow events that generate intense dust storms (haboobs).

Line 44-45, autogenic vs. allogenic processes, not clear what is specifically referred to in this paper. It will be helpful for readers if the terms in the subsequent sentences are linked directly to these two terms.

This wording has been removed.

Line 64-67, this sentence is correct but does not fit in here well for the writing does not extend to explain why studying past dust record is relevant to changes induced by human activities.

We have added clarification on how long records are relevant context to human driven changes (lines 29-30)

Line 73, rapid climate change only refers to glacial/interglacial cycles here, be specific.

This has been removed.

Line 133, the 5 end members are not described.

The five end members are now described in detail in the Results (lines 79-84)

Line 138, It is confusing to lump quartz with a clay mineral and a heavy mineral zircon together. Rephrase to something as quartz accounts XX% (what is the link between quartz and basalt?), illite is the dominant clay mineral and it can not produced from weathering basalt, and abundant zircon grains can not be from basalt.

This section has been rewritten for clarity (lines 68-74).

Line 140, the connection between the two sentences is not clear. How do quartz, zircon and illite link to EM3 ad Em5, are they mostly in this two end members? No evidence is provided.

We have clarified the mineralogical differences between lake sediments and local bedrock earlier on in the Results section (lines 68-74). We then use geochemical stratigraphy (contextualized by its relationship to mineralogical observations) to correlate with stratigraphic trends in end member abundance: Rb representing illite (fine dust – EM3), Zr representing zircon (coarse dust - EM4), and Ti representing ilmenite (local alluvium - EM1+EM2+EM5). These geochemical indicators are used rather than direct mineralogical stratigraphy because our mineralogical analyses do not have sufficient resolution to compare directly to end member abundances. Geochemical plots previously in the supplemental has been moved into Figure 3 to make these arguments clearer.

Line 142-143, how about flow velocity? Strong vs. weaker wind.

The discussion on what the two grain size populations of dust represent, either a difference in source distance or difference in mode/strength of transport has been moved, appropriately, to the discussion. According to the physics of settling dust particles discussed in Tsoar and Pye (1987), particles larger than 20 μm are unlikely to travel more than tens of kilometers, consistent with our interpretation that EM4 (38 μm) represents a nearby source. While an intense storm such as a haboob may pick up greater numbers of large particles, these events are relatively short-lived (minutes to hours) and dust grains $>20 \mu\text{m}$ quickly settle out. Particles $<20 \mu\text{m}$, however, are capable of traveling 100's of kilometers, consistent with the interpretation that fine dust (EM3) is regionally sourced.

Line 181-190, why EM2 can not be dust with sources between those of EM3 and EM4? What is the relationship between EM5 and $Ti/(Zr+Rb)$? EM2 peak is high, what about the element ratio is mainly driven by EM2? Also can EM2 be the weathering product of EM3 and EM4 (thus ultimately dust) given that catchment soils have high EM3-5? Can EM5 be from episodic sandstorms?

EM2 (5 μm) is smaller than EM3 (13 μm) and EM4 (38 μm), so it doesn't make sense as an intermediate source.

All local clastic sediments likely contain ilmenite as it is the most resistant mineral present within the local bedrock. Because most Ti in this system is likely derived from ilmenite (Staley et al., 2023), Ti is representative of EM1+EM2+EM5.

EM2's existence as an end member implies that its deposition trends are distinct. For instance, if EM2 were a weathering product of EM3 or EM4, it would be deposited contemporaneously and exhibit the same stratigraphic variability of those end members. Therefore, the end member model would not identify it as a distinct grain size population and instead lump it in with one of those end members that would then either have a broader particle size distribution or exhibit two distinct modes, the dust and the weathering product. Case in point, EM4 has a left shoulder in the clay to fine silt range (Fig. 2). The model inherently lumps the main modal peak and this shoulder because they covary in the stratigraphy, suggesting that their deposition is controlled by the same process. Therefore, the shoulder is interpreted as former particle aggregates that were disaggregated post-deposition, either from weathering, shallow diagenesis, or sample processing.

EM5 is highly unlikely to be related to dust storms. EM5 has a main modal peak of 169 μm , fine sand. While fine and even medium sand has been documented as suspended load in intense dust storms, these grain sizes represent the maximum and are unlikely to define the sandstorm's modal grain size distribution.

Figure 3 has lots of information not mentioned in the text.

We undertook a major reorganization of figures to narrow their focus. Some elements of this figure have been either removed (LR04) or moved to another figure (erosion rates). Figure 4 maintains the straightforward comparison between the Stoneman Lake total dust record and other dust records from the Northern Hemisphere.

Line 207, Tau et al say that “Particle-size distributions reveal fine dust transported during winter from the northwestern Sonoran Desert and the Mojave Desert and coarse dust transported during summer from the southwestern Sonoran Desert, similar to current climate systems and dust pathways.” Different from the argument that is cited for.

This has been corrected. While we agree with Tau et al.’s particle size results and interpretation that there are two distinct dust populations in their record, we disagree that dust end member size distributions can be directly linked to particle size distributions of dust sources and therefore atmospheric circulation modes (Tau et al. 2021 Fig 2). All particle size distributions of their potential sources contain significant amounts of particles that could fit into either of their dust end members. Source particle size distributions only define what can be contributed to initial emissions, not what gets deposited after 100’s of kilometers of transport and gravitational sorting. Furthermore, the distances between their proposed winter and summer sources are small; it is unlikely that this small difference results in such distinct dust end member sizes.

Line 220, Even the westerlies are stronger during glacier period, how about monsoon? McGee et al. say “We demonstrate that a wide range of data supports wind gustiness as a primary driver of global dust levels”, which is exactly opposite to what the paper is cited to support.

This has been corrected and the reference has been removed. We note that McGee et al. does not consider or incorporate data from North America in their study. As we show in this paper, the dust system in western North America differs clearly from the rest of the globe. We also clarify the glacial-interglacial variability in monsoon strength (lines 299-301).

Line 221, do we know roughness should be higher or lower during glacial period?

Based on the paleoecological study of Stoneman Lake by Jiménez-Moreno et al. (2023), we know there was a denser forest canopy around the lake during glacials (lines 311-313).

Line 233, appear to be influenced by global climate changes, is it specifically glacier-interglacial cycles? The reasoning flow is confusing here.

This specific statement has been removed and the discussion of global glacial-interglacial climate change and how it impacts dust variability in southwestern North America versus elsewhere has been completely rewritten for clarity.

Line 245, “wetter”, you said monsoon and tropical cyclones were suppressed, how come it became wetter or have increased effective moisture (what is that)?

This has now been clarified (lines 174-177). Compared to classic monsoonal regimes, the North American Monsoon is relatively weak and not the primary driver of long-term effective moisture (Precipitation – Evaporation) variability in southwestern North America. Here, the main drivers of long-term trends are temperature and precipitation changes resulting from glacial cycles. Cooler temperatures and the steering of westerly-derived moisture by the Laurentide Ice Sheet increase effective moisture causing pluvial conditions during glacials. When the ice sheet retreats during interglacials, this steering effect is removed, and temperatures increase leading to drier conditions overall, despite increased incidence of monsoons and cyclones.

Line 295, “Verde Valley dust” is there any geochemical evidence to support the interpretation? This is an important component of the dust generation mechanism.

Unfortunately not, although we have discussed some projects to examine this. In the place of geochemical correlation of source and sink, we support the interpretation of EM4 as proximally derived dust using particle size dependence on transport distance, i.e., the majority of particles >20 μ m are unlikely to travel more than 10's of kilometers, and temporal correlation between depositional periods in the Verde Valley and increased coarse dust accumulation (lines 220-231, Fig. 6).

Author's Response - NCOMMS-24-02264C

Comments are italicized and our responses are in regular font.

Comments from Reviewer #2:

This manuscript has undergone a significant transformation. My perception of the work and its significance has completely changed after reading this version.

The work presents a long and high-resolution dust record from southwestern North America, providing valuable insights into the complex interactions between dust generation, climate, vegetation, and erosion processes. This research is important not only for the last 200,000 years but also has implications for the study of older loess, given that loess records extend further back into geological history. The manuscript is well-written, exhibiting a logical flow of reasoning. The figures are well-crafted and enhance the reader's understanding. The methods section is sufficiently detailed, allowing readers to evaluate the uncertainty associated with the end-member analysis. Furthermore, the conclusions and interpretations are well-supported by the data. I am very impressed with the scientific narrative and the authors' ability to convey the story. I thoroughly enjoyed reading this work and do not have any major comments, just a few minor suggestions to enhance clarity. While I hold a very positive view of this study, I would like to note that I am not a Quaternary geologist and may lack the expertise to fully evaluate the specifics of Quaternary regional climate.

1. In Figure 4, MIS 3 is indicated in light blue, please add a note in the figure caption to explain this distinction. Additionally, the three proxies from the Chinese Loess Plateau, Japan, and the Northwest Pacific appear to be primarily influenced by dust production in Central Asia, raising questions about why this is considered a global flux. The dust curve from the East Atlantic shows significant differences compared to the other three proxies. It may be worthwhile to adjust the text in the related area, replacing "global" with "other areas around the world," as the main point is that Southwest North America differs from these other regions concerning the causal relationship between climate and dustiness.

To simplify, we have removed the light blue indicator for MIS 3 in Figures 3, 4, and 6 and instead only indicate the strongest glacial periods in darker blue (MIS 2, 4, 6). To clarify, "Global DMARs" has been changed to "Other Northern Hemisphere records of DMAR" within Figure 4.

2. In Figure 6, there are several climate proxies included. Given that the text may not have enough space to explain how each proxy indicates different climate conditions to support the interpretation, it would be helpful to annotate the figure to indicate warmer/cooler and drier/wetter conditions, particularly for $\delta^{18}\text{O}$. Additionally, since the text discusses 20 kyr cycles, marking these cycles in the figure would enhance clarity.

Arrows have been added to $\delta^{18}\text{O}$ curves (**b**, **c**) to indicate wetter vs drier conditions. I have also added arrows to **i**, the charcoal record, and **k**, the relative lake depth record. Within the figure caption I have noted that MIS 5 and 7 substage boundaries generally align with 20-kyr precession cycles.

Comments from Reviewer #4:

The Stoneman Lake core has an impressive ~230 kyr record of dust that has allowed Staley et al. to consider controls on glacial-interglacial timescales of dust flux. The conclusion that interglacials are dustier than glacials is an interesting one, and they link the dynamics of the eolian system back to climatic-driven changes in sediment supply. Tau et al. (2021) made a similar conclusion for Holocene dust at nearby Montezuma Well, and so it is nice to see some broad agreement, although the two papers differ on atmospheric controls and where the dust may have come from. I also think that geomorphologists would expect that in some places, regional controls may counter global trends in dust emissions, and there is a strong case for that here. Their end member modelling identified two distinct dust modes (EM3 and EM4). They found that the EM3 and EM4 modes are not always synced, suggesting a heterogeneous landscape response to climate change and dust emissions. I like this because regional dust sources are not controlled by an on-off switch, as is assumed in some dust emission models. The long geological record of dust as well as results that counter global trends in dust are significant contributions and worthy of publication in Nature Communications.

I think the paper reads well – it gets to the point and provides the key data to support their conclusion. This is my first review and having seen the author responses to the previous reviewer comments, I sense there have been a lot of improvements in both the writing and figures. I can follow the figures that support their interpretations.

I think the models and reasoning they present to explain the different grain size populations as identified though end member modeling make sense, although I think some clarification is needed for EM4 coarse dust sources. The authors suggest that the primary source of EM4 is the Verde Valley. They have identified the location of deposits on Fig. 1. They include several citations of maps and reports documenting these deposits (citations 54-60). However, they make an assumption that the Verde River deposits were a source of dust. Reviewer 2 posed a similar concern, and the authors replied that no geochemical provenance analysis has been done to prove the linkage. Their assumption, and their model, could falter if the river was gravel dominated in this location because they present no alternative source. Significant fine grained deposits are required to warrant wind erosion and dust production. Do citations 54-60 identify fine grained fluvial deposits? If so, this is not stated directly. The authors seem to assume the river would “periodically exposing fresh fine sediments to the atmosphere” without providing evidence of the fine sediment. If 54-60 document such fine deposits, say so directly. If not, or in addition to, consider checking the soil survey for this area. I was able to see the area highlighted in yellow on Fig 1 in soilweb-apps, where some of the area is dominated by sandy loams and other fine soils at the surface. Confirming the existence of fine grained (sandy) deposits makes their model seem plausible.

This is an important aspect to clarify. After reviewing the contained references (#’s 54-60) and available soil survey data (AZ639, AZ641, and AZ643 surveys), we have confirmed that nearly every fluvial and alluvial deposit in the Verde Valley contains at least some fine-grained sediment (line 229). We have also noted the Chemehuevi Formation’s fine-grained nature (line 244).